# Clinical impact of pharmacogenetic risk variants in a large chinese cohort

Incorporating pharmacogenetics into clinical practice promises to improve therapeutic outcomes by optimizing drug selection and dosage based on genetic factors affecting drug response. A key advantage of PGx-guided therapy is to decrease the likelihood of adverse events. To evaluate the clinical impact of PGx risk variants, we performed a retrospective study using genetic and clinical data from the largest Han Chinese cohort, comprising 486,956 individuals, assembled by the Taiwan Precision Medicine Initiative. We found that nearly all participants carried at least one genetic variant that could affect drug response, with many carrying multiple risk variants. Here we show the detailed analyses of four gene-drug pairs, azathioprine (*NUDT15*/*TPMT*), clopidogrel (*CYP2C19*), statins (*ABCG2*/*CYP2C9*/*SLCO1B1*), and NSAIDs (*CYP2C9*), for which sufficient data exists for statistical power. While the results validate previous findings that PGx risk variants are significantly associated with drug-related adverse events or ineffectiveness, the excess risk of adverse events or lack of efficacy is small compared to that found in those without the PGx risk variants, and most patients with PGx variants do not suffer from adverse events. Our results point to the complexity of implementing PGx in clinical practice and the need for integrative approaches to optimize precision medicine.

Variability in drug effectiveness and/or safety greatly impacts therapeutic outcome, with drug-response rates varying widely from 25% to 80% among the commonly used drugs[1]. Twin and case/control studies have established that genetic factors contribute to variability in drug response. Pharmacogenetics (PGx) explores the influence of an individual's genetic makeup on drug metabolism, efficacy, and adverse events[2]. Incorporating PGx into clinical practice is a promising strategy for healthcare clinicians to tailor drug selection and dosing to maximize therapeutic benefits while minimizing the risk of drug-related adverse events[2,3]. PGx also gains increasing attention in pharmaceutical industry for its potential in drug development and drug repurposing[4].

To facilitate the clinical implementation of PGx, experts from regulatory agencies and consortia such as the United States Food and Drug Administration (US FDA)[5,6] and Clinical Pharmacogenetics

Implementation Consortium (CPIC)[7] have published clinical practice recommendations based on level of evidence and advocated for incorporating pharmacogenetic testing when a relevant drug is being prescribed. In a comprehensive study examining PGx variation within the UK Biobank, it was found that the average participant carried genetic variants that would affect their response to around 10 drugs based on CPIC guidelines[8]. In Australia, merely 4% of the study participants lacked actionable PGx variants, and 42% of them had more than 2 actionable PGx variants[9]. These findings argue for large-scale implementation of PGx-guided therapy. However, the discordant recommendations from different agencies and consortia make clinical implementation of PGx challenging[10–12]. Furthermore, the benefits of PGx-guided therapy have not been established in large, population-based studies, especially in non-European populations. Therefore, it is crucial to perform studies of

✉ e-mail: nancy.wei324@gmail.com; chchen@ibms.sinica.edu.tw; chiencmail@gmail.com; a00001@mail.fjuh.fju.edu.tw; yenling.chiu@gmail.com; alex0624x@yahoo.com.tw; Pui.Kwok@ucsf.edu

large cohorts to evaluate the influences of PGx variants before clinical implementation.

The Taiwan Precision Medicine Initiative (TPMI), a consortium of researchers from the Academia Sinica and 33 partner hospitals across Taiwan, has enrolled 486,956 participants and obtained genetic and longitudinal clinical data from each person[13]. With access to their drug prescription and drug-related adverse event history on several commonly prescribed drugs, we conducted a retrospective study to analyze four PGx gene-drug pairs with dosing recommendations from CPIC and US FDA to determine the impact of PGx risk variants on drug response and toxicity in the largest Asian cohort ever studied in PGx. Specifically, we evaluated the association between genetic variants and risk for adverse events, including *NUDT15/TPMT* and azathioprine (AZA)-induced myelosuppression, *CYP2C19* and clopidogrel-related major adverse cardiovascular events (MACEs), *ABCG2/CYP2C9/SLCO1B1* and statin-associated myopathy (SAMs), and *CYP2C9* and non-steroidal anti-inflammatory drugs (NSAID)-linked gastrointestinal (GI) and renal toxicity. These pairs were selected based on sufficient sample size for statistical power, and the availability of relevant clinical data. To ensure both clinical relevance and robust methodological design, we focused on gene-drug pairs that could be reliably analyzed. As discussed below, limitations in CYP2D6 genotyping meant that several important pairs could not be analyzed and were excluded from the study.

## Results

### Landscape of clinically actionable pharmacogene variants in the cohort

The 486,956 Han Chinese participants of the TPMI were genotyped with one of two SNP arrays (TPMv1 with 686,463 SNPs or TPMv2 with 743,227 SNPs) that contained 3949 and 2911 PGx markers, respectively. In this study, we extracted for the cohort the risk variant (star alleles) status of clinically actionable PGx markers or human leukocyte antigen (*HLA*) types in 19 pharmacogenes together with the associated phenotypes, including their effects on metabolic enzymes, transporters, immune mediators, and mitochondria proteins (Supplementary Tables 1, 2, and Fig. 1a). Variants in these 19 genes affected the response of 58 commonly prescribed drugs. Overall, 99.9% of TPMI participants possessed at least one PGx variant, which was mainly due to the highly prevalent *VKORC1* rs9923231 (−1639 G > A) variant, as previously seen in other Han Chinese cohorts[14,15]. On average, each TPMI participant carried 4.3 clinically actionable PGx risk variants (Fig. 1b).

### Drug use in people carrying PGx

We extracted and analyzed the drug prescription data from the electronic medical record (EMR) of TPMI participants to determine their drug usage. Among the TPMI participants, 48.7% of the TPMI participants have been prescribed at least one of the 58 drugs with clinical practice recommendations based on their genetic status in the 19 pharmacogenes. Additionally, 28.4% took two or more drugs with PGx information (Supplementary Table 3). Individuals with *CYP2C19* loss-of-function (LoF) alleles and *SLCO1B1* decreased or poor function alleles have been exposed to more drugs with clinical practice recommendations. Among the individuals carrying clinically actionable PGx variants, 17.8% of them have been prescribed the responding high-risk drugs, defined as those requiring dose adjustments, alternative therapies, or additional monitoring when prescribed to individuals with actionable PGx variants (Fig. 2 and Supplementary Table 2). The top 20 most commonly prescribed drugs included statins, NSAIDs, proton-pump inhibitors (PPI), anti-platelet drugs, and antibiotics (Table 1).

Based on the PGx clinical practice recommendations from US FDA and CPIC, those carrying actionable PGx variants needed to have adjusted dosage, to be given alternative drugs, or to be paid extra attention when they were prescribed with high-risk drugs (Table 1). Because the PGx risk variant status of these individuals was unknown when they were prescribed the drugs, the PGx clinical practice recommendations were not followed. However, based on the available medical data, most patients did not suffer from any of the predicted adverse events. For instance, 85.9% of *CYP2C19* LoF allele carriers tolerated clopidogrel without MACE, 78–83% of *NUDT15/TPMT* risk allele carriers tolerated azathioprine, and over 98% of individuals with high-risk statin-related PGx profiles did not develop muscle-related adverse events.

### Treatment outcomes of those with PGx risk variants/*HLA* types

We examined the clinical outcomes of TPMI participants who were prescribed drugs known to be affected by PGx variants/*HLA* types by assessing their EMRs for the drug prescribed, any dosage/drug changes or clinical intervention, and lab test results. By comparing the therapeutic outcomes of patients with or without clinically actionable PGx variants/*HLA* types, we aimed to determine the impact of PGx screening in reducing adverse drug events (ADEs) and improving therapeutic responses. We selected 4 gene-drug pairs for which the TPMI data had large enough sample size and comprehensive clinical data for outcome assessment.

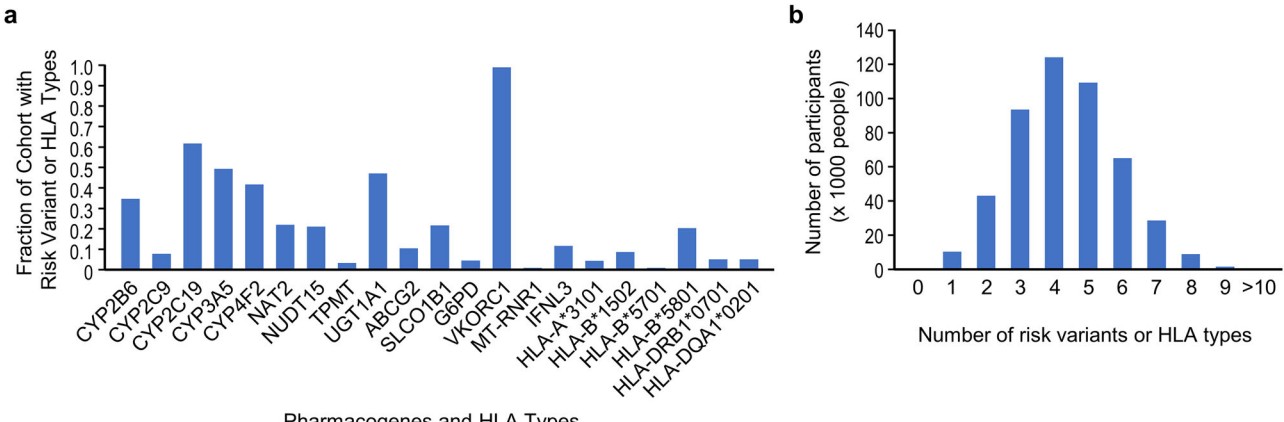

**Fig. 1 | TPMI participants with actionable variants/*HLA* types in 19 pharmacogenes. a** The fraction of TPMI participants with actionable variants/*HLA* types in pharmacogenes; (**b**) The number of individuals carrying actionable PGx risk variants or *HLA* types.

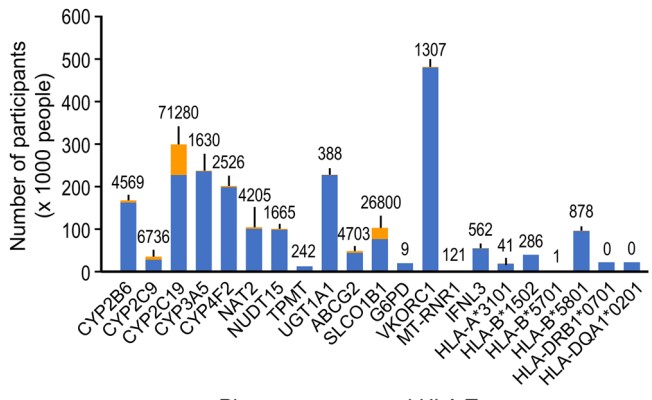

**Fig. 2 | The distribution of risk drugs prescription in people carrying actionable PGx variants.** The number of people carrying actionable PGx risk variants or *HLA* types who took (orange) or did not take (blue) the drug for which they were at risk; number above bar denotes those who took the drug for which they were at risk.

## Impact of *CYP2C19* risk variants on clopidogrel-related MACE

Clopidogrel is an anti-platelet drug commonly used in patients with unstable angina (UA), myocardial infarction (MI), stroke, and peripheral arterial disease to prevent MACE. Clopidogrel is a prodrug that requires enzymatic activation, primarily through *CYP2C19*. A meta-analysis showed that clopidogrel exhibited a significantly higher risk of MACE in patients with *CYP2C19* LOF alleles[16].

Data from 28,055 clopidogrel users who met the criteria for inclusion were used in this study (Supplementary Fig. 1). Among them, 12.4% developed MACE, including MI (2.1%), UA (2.5%), heart failure (HF, 4.7%), target lesion revascularization (TLR, 3.7%), stroke (2.3%), or cardiovascular death (CV death, 0.4%). The demographics, clinical characteristics, and *CYP2C19* status of the patients were found in Supplementary Table 4. There was no significant difference in age, gender, and comorbidities between individuals with or without MACE. Of note, concomitant use of the proton pump inhibitors showed significant association with clopidogrel-related MACE (35.3% vs. 40.8%, $P = 1.6 \times 10^{-10}$), which aligned with previous findings from a nationwide population-based study using the Taiwan National Health Insurance database[17].

Patients taking clopidogrel with either one or two *CYP2C19* LOF alleles were significantly associated with MACE compared to those with no LoF alleles ($P = 2.97 \times 10^{-27}$, OR = 1.53, 95% CI = 1.42–1.65), including MI ($P = 2.09 \times 10^{-31}$, OR = 3.98, 95% CI = 3.15–5.02), UA ($P = 7.24 \times 10^{-13}$, OR = 1.88, 95% CI = 1.58–2.23), HF ($P = 1.77 \times 10^{-4}$, OR = 1.37, 95% CI = 1.22–1.55), TVR ($P = 1.91 \times 10^{-4}$, OR = 1.29, 95% CI = 1.13–1.47), and stroke ($P = 6.31 \times 10^{-5}$, OR = 1.41, 95% CI = 1.19–1.67) under multivariate analysis (Fig. 3a and Supplementary Table 5). Both *CYP2C19* intermediate metabolizers (IM) and poor metabolizers (PM) had higher MACE incidence under clopidogrel treatment; however, 85.9% of people with *CYP2C19* LOF alleles tolerated clopidogrel treatment well. There was no significant difference in risk of CV death under clopidogrel therapy for those with or without *CYP2C19* LoF alleles (Fig. 3b, c and Supplementary Table 5).

## Impact of *NUDT15*/*TPMT* in AZA-related adverse events

AZA is a commonly prescribed immunosuppressive antimetabolite in the management of acute lymphoblastic leukemia, autoimmune conditions, and organ transplantation. AZA has a narrow therapeutic index and a high potential for ADEs, like bone marrow toxicity, hepatotoxicity, massive hair loss, nausea and vomiting. Nudix hydrolase 15 (*NUDT15*) and thiopurine-S-methyltransferase (*TPMT*) are two important enzymes that decrease the concentration of AZA active

metabolites, so higher AZA toxicity risk is commonly observed in people with deficient *NUDT15* and *TPMT*[18–20]. Both CPIC and US FDA recommend a significant or substantial dose reduction for *NUDT15* or *TPMT* IM and PM[6,21].

We studied 8451 participants using AZA for prevention of renal transplant rejection and treatment of inflammatory conditions, including systemic lupus erythematosus (SLE), Sjogren syndrome, rheumatoid arthritis (RA), other connective tissue disorders, Crohn's disease, ulcerative, or severe atopic dermatitis (Supplementary Fig. 2). A total of 1503 (17.8%) patients stopped AZA treatment due to intolerable ADEs such as leukopenia (10.7%), thrombocytopenia (6.8%), hepatitis (4.6%), GI discomfort (1%), alopecia (0.7%), and allergy (0.6%). The distribution of sex, comorbidity, and usage of concurrent drugs (aspirin, allopurinol, corticosteroids, and methotrexate) were not significantly different in the patients with AZA-induced ADEs compared with AZA tolerant controls, but concomitant use of allopurinol and AZA increased risk for adverse events development (Supplementary Table 6). The age of initial use of AZA in the ADE group was slightly younger than the tolerant control group.

The *NUDT15* IM ($P = 5.44 \times 10^{-4}$, OR = 1.28, 95% CI = 1.11–1.48) and *NUDT15* PM ($P = 2.88 \times 10^{-8}$, OR = 4.05, 95% CI = 2.47–6.64) phenotypes were associated with AZA-induced adverse events under multivariant regression analysis (Fig. 4a–c and Supplementary Table 7). The incidences of leukopenia in intermediate/poor *NUDT15* metabolizers (14.9% and 37.7%, respectively) patients were significantly higher than that in extensive *NUDT15* metabolizers (NM; 9.4%) ($P = 4.47 \times 10^{-14}$, OR = 1.85, 95% CI = 1.58–2.17). The significant associations between AZA discontinuation due to thrombocytopenia ($P = 4.91 \times 10^{-3}$, OR = 2.67, 95% CI = 1.35–5.3) or massive hair loss ($P = 1.24 \times 10^{-5}$, OR = 10.84, 95% CI = 3.72–31.58) and *NUDT15* were only observed in PMs. AZA discontinuation rates due to adverse events did not differ between patients with normal and decreased *TPMT* activity (Fig. 4d–e and Supplementary Table 7).

Again, the clinical impact was limited, with only 21.4% of *NUDT15* IM/PM and 16.7% of *TPMT* IM having severe AZA-induced ADEs, whereas 16.9% of *NUDT15* NM and 17.8% of *TPMT* NM suffered from AZA-induced ADEs.

## Impact of *ABCG2*/*CYP2C9*/*SLCO1B1* in SAMs

Statins are used in reducing plasma low-density lipoprotein cholesterol (LDL-C) levels and preventing the risk of atherosclerotic cardiovascular diseases[22]. However, statins use is often associated with side-effects, particularly those related to musculoskeletal and hepatic systems. SAMs stand out as the most frequently reported adverse events, encompassing a spectrum of clinical presentations ranging from mild symptoms, such as muscle pain and weakness, to severe muscle injury, such as rhabdomyolysis[23]. Establishing the causal relationship between muscle complaints and statin use is challenging, particularly in the case of subjective symptoms such as myalgia[24,25]. To mitigate the risk of SAMs over-diagnosis, we reviewed and extracted EMR data on symptom relief following statin withdrawal or symptom recurrence upon statin rechallenge. Data from 127,197 TPMI participants who ever treated with any statin agent (atorvastatin, fluvastatin, lovastatin, pitavastatin, pravastatin, rosuvastatin, or simvastatin) were analyzed for this study (with individuals previously suffering from muscular disorders excluded). Of these participants, 34,411 participants took more than one statin agent, and 7 were treated with 6 different statin agents (Supplementary Table 8). The switching between statins in these rare cases was attributed to reasons such as intolerance and inefficacy. Specifically, these seven patients experienced adverse events with certain statins, such as muscle toxicity, prompting switches to alternative statins. In some cases, the next statin was well-tolerated but failed to adequately control LDL levels, necessitating further changes. Ultimately, all seven patients were transitioned to non-statin therapies, including restricted diet control. Although statins

**Table 1 | Clinical practice recommendations for the TPMI participants who took the drug based on their PGx phenotype**

| High-risk drugs | Total | Standard treatment | Increased dose | Decreased dose | Alternative | Enzyme activity test | Used in caution | Indetermined |
|---|---|---|---|---|---|---|---|---|
| Atorvastatin | 70,273 | 55,516 | | 14,752 | | | | 5 |
| Celecoxib | 50,200 | 46,570 | | 3608 | | | | 22 |
| Rosuvastatin | 46,617 | 17,019 | | 5263 | | | 24,320 | 15 |
| Lansoprazole | 44,378 | 17,063 | 284 | 26,955 | | | | 76 |
| Clopidogrel | 33,664 | 13,101 | | 20,517 | | | | 46 |
| Omeprazole | 33,407 | 12,807 | 210 | 20,344 | | | | 46 |
| Pitavastatin | 30,876 | 24,284 | | 6591 | | | | 1 |
| Flurbiprofen | 28,989 | 26,807 | | 2159 | | | | 23 |
| Pantoprazole | 28,138 | 10,889 | 185 | 17,024 | | | | 40 |
| Gentamicin | 23,953 | 23,816 | | | 109 | | | 28 |
| Dexlansoprazole | 19,147 | 7387 | | 11,604 | 122 | | | 34 |
| Piroxicam | 13,933 | 12,841 | | 1061 | 17 | | | 14 |
| Simvastatin | 10,230 | 7973 | | 2256 | | | | 1 |
| Meloxicam | 10,174 | 9447 | | 715 | 7 | | | 5 |
| Sulfasalazine | 9777 | 7679 | | | | | 2,082 | 16 |
| Imipramine | 9636 | 8195 | | | 1420 | | | 21 |
| Azathioprine | 8486 | 6602 | | 1774 | 72 | | | 38 |
| Sulfamethoxazole and Trimethoprim | 7699 | 6016 | | | | | 1666 | 17 |
| Pravastatin | 6955 | 5510 | | 1445 | | | | - |
| Citalopram | 6942 | 2714 | | 4160 | 51 | | | 17 |
| Escitalopram | 6845 | 2677 | | 4100 | 51 | | | 17 |
| Doxepin | 6773 | 5745 | | | 1018 | | | 10 |
| Ibuprofen | 6681 | 6135 | | 541 | | | | 5 |
| Warfarin | 6111 | 1203 | | 4905 | | | | 3 |
| Allopurinol | 5251 | 4373 | | | 878 | | | |
| Sertraline | 4786 | 3895 | | 868 | 12 | | | 11 |
| Tobramycin | 4190 | 4168 | | | 16 | | | 6 |
| Lovastatin | 3563 | 2784 | | 779 | | | | - |
| Fluvastatin | 3526 | 2548 | | 968 | 5 | | | 5 |
| Tacrolimus | 3299 | 1668 | 1630 | | | | | 1 |
| Isoniazid | 2918 | 2269 | | | | | 646 | 3 |
| Oxcarbazepine | 2229 | 2056 | | | 173 | | | |
| Ribavirin | 1873 | 1591 | | | | | 282 | |
| Carbamazepine | 1319 | 1210 | | | 109 | | | |
| Clobazam | 1283 | 484 | | 799 | | | | - |
| Abacavir | 692 | 691 | | | 1 | | | |
| Phenytoin | 670 | 574 | | 42 | 54 | | | - |
| Amikacin | 635 | 631 | | | 3 | | | 1 |
| Irinotecan | 596 | 324 | | 264 | | | | 8 |
| Peginterferon alfa-2a | 576 | 504 | | | | | 72 | |
| Voriconazole | 343 | 292 | | | 51 | | | |
| Efavirenz | 258 | 179 | | 79 | | | | - |
| Nateglinide | 227 | 227 | | - | | | | - |
| Peginterferon alfa-2b | 218 | 181 | | | | | 37 | |
| Pazopanib | 120 | 120 | | | | | | |
| Dapsone | 108 | 96 | | | 2 | 5 | | 5 |
| Nilotinib | 73 | 66 | | | | | 6 | 1 |
| Streptomycin | 59 | 59 | | | - | | | - |
| Atazanavir | 56 | 50 | | | 6 | | | - |
| Mercaptopurine | 37 | 30 | | 6 | - | | | 1 |
| Rasburicase | 31 | 29 | | | 1 | - | | 1 |
| Brivaracetam | 28 | 25 | | 3 | | | | - |
| Lapatinib | 27 | 27 | | | | | | |
| Tenoxicam | 23 | 23 | | - | - | | | - |
| Paromomycin | 15 | 15 | | | - | | | - |
| Abrocitinib | 11 | 9 | | 2 | | | | |
| Nitrofurantoin | 5 | 4 | | | - | 1 | | - |
| Kanamycin | 5 | 5 | | | - | | | - |

## a. *CYP2C19* LOF

| Outcome | Tolerant(NM) | Event(NM) | Tolerant(LOF) | Event(LOF) | OR(95% CI) | P-value |
|---|---|---|---|---|---|---|
| MACE | 9,856 | 1,062 | 14,687 | 2,411 | 1.53(1.42-1.65) | 2.96E-27 |
| MI | 10,834 | 84 | 16,583 | 515 | 3.98(3.15-5.02) | 2.09E-31 |
| UA | 10,737 | 181 | 16,574 | 524 | 1.88(1.58-2.23) | 7.24E-13 |
| HF | 10,497 | 421 | 16,205 | 893 | 1.37(1.22-1.55) | 1.77E-07 |
| TLR | 10,574 | 344 | 16,408 | 690 | 1.29(1.13-1.47) | 1.91E-04 |
| CV_death | 10,883 | 35 | 17,028 | 70 | 1.25(0.82-1.89) | 0.30 |
| Stroke | 10,712 | 206 | 16,650 | 448 | 1.41(1.19-1.67) | 6.31E-05 |

## b. *CYP2C19* IM

| Outcome | Tolerant(NM) | Event(NM) | Tolerant(IM) | Event(IM) | OR(95% CI) | P-value |
|---|---|---|---|---|---|---|
| MACE | 9,856 | 1,062 | 11,245 | 1,811 | 1.50(1.39-1.63) | 6.76E-23 |
| MI | 10,834 | 84 | 12,668 | 388 | 3.90(3.08-4.95) | 2.75E-29 |
| UA | 10,737 | 181 | 12,665 | 391 | 1.84(1.53-2.20) | 3.38E-11 |
| HF | 10,497 | 421 | 12,392 | 664 | 1.34(1.18-1.52) | 5.27E-06 |
| TLR | 10,574 | 344 | 12,536 | 520 | 1.27(1.11-1.46) | 7.28E-04 |
| CV_death | 10,883 | 35 | 13,004 | 52 | 1.19(0.76-1.84) | 0.45 |
| Stroke | 10,712 | 206 | 12,718 | 338 | 1.40(1.17-1.67) | 2.11E-04 |

## c. *CYP2C19* PM

| Outcome | Tolerant(NM) | Tolerant(PM) | Event(NM) | Event(PM) | OR(95% CI) | P-value |
|---|---|---|---|---|---|---|
| MACE | 9,856 | 3,442 | 1,062 | 600 | 1.62(1.46-1.81) | 2.05E-18 |
| MI | 10,834 | 3,915 | 84 | 127 | 4.22(3.19-5.57) | 3.87E-24 |
| UA | 10,737 | 3,909 | 181 | 133 | 2.02(1.60-2.54) | 1.90E-09 |
| HF | 10,497 | 3,813 | 421 | 229 | 1.49(1.26-1.76) | 3.08E-06 |
| TLR | 10,574 | 3,872 | 344 | 170 | 1.34(1.11-1.62) | 2.72E-03 |
| CV_death | 10,883 | 4,024 | 35 | 18 | 1.44(0.81-2.55) | 0.21 |
| Stroke | 10,712 | 3,932 | 206 | 110 | 1.46(1.15-1.85) | 1.89E-03 |

**Fig. 3 | Impact of *CYP2C19* in clopidogrel-related MACE.** Forest plot of the MACE risk in clopidogrel users with different *CYP2C19* phenotypes: (**a**) people who carried at least one *CYP2C19* LoF alleles vs non-carriers, (**b**) *CYP2C19* IM vs NM, and (**c**) *CYP2C19* PM vs NM. The case number, OR, 95% CI, and *P* value were listed in the table. The ORs and 95% CIs were estimated using logistic regression adjusting for covariates (two-sided test). Data points represent odds ratios; error bars represent 95% confidence intervals. Significant associations are shown in red. CI = confident interval; CV_death = cardiovascular death; HF = heart failure; IM = intermediate metabolizer; LoF = loss-of-function; MACE = major adverse cardiovascular events; MI = myocardial infarction; NM = normal metabolizer; OR = odds ratio; PM = poor metabolizer; TLR = target lesion revascularization; UA = unstable angina.

shared similar structures, many participants who suffered from muscle toxicity caused by one statin could take another statin without any adverse events (Supplementary Table 8). Therefore, genetic associations with myopathy induced by different statins were analyzed separately (Fig. 5 and Supplementary Table 9).

Our study showed that people with poor function of *ABCG2* had higher risk of myositis when taking atorvastatin (P = $3.58 \times 10^{-3}$, OR = 2.27, 95% CI = 1.31–3.94) or simvastatin (P = $1.26 \times 10^{-2}$, OR = 3.43, 95% CI = 1.3–9.05); whereas people with *ABCG2* decreased function phenotype had higher risk of myalgia while taking fluvastatin (P = $5.39 \times 10^{-3}$, OR = 1.97, 95% CI = 1.22–3.18) (Fig. 5a–c and Supplementary Table 9). Patients with *SLCO1B1* decreased or poor function phenotype (c.521 T > C) had increased risk in developing myalgia with atorvastatin (P = $3.35 \times 10^{-3}$, OR = 1.27, 95% CI = 1.08–1.48) and severe SAM with simvastatin (P = $3.14 \times 10^{-3}$, OR = 2.91, 95% CI = 1.43–5.91) (Fig. 5d, e and Supplementary Table 9). Furthermore, although the pharmacokinetic of fluvastatin was known to be affected by *CYP2C9* phenotypes, the frequency of *CYP2C9* LOF was not significantly higher in patients who experienced SAM after receiving fluvastatin in our cohort (P = $8.63 \times 10^{-1}$, OR = 0.93, 95% CI = 1.22–3.18) (Fig. 5f and Supplementary Table 9).

Despite the significant association in some of the ADEs, the frequencies were so low that the vast majority of the "high risk" individuals (98% or more) did not suffer from any ADEs.

### Impact of *CYP2C9* in NSAID-associated adverse events

NSAIDs are known for their ability to reduce pain, inflammation, and fever by inhibiting the production of prostaglandins. Although substantial evidence links *CYP2C9* deficient phenotype to altered NSAID plasma concentrations, clinical evidence directly substantiating an increased risk of ADEs in individuals with reduced *CYP2C9* metabolism of NSAIDs (such as celecoxib, flurbiprofen, ibuprofen, lornoxicam, meloxicam, piroxicam and tenoxicam) remains limited[26]. However, since the NSAID toxicity is dose- and duration-dependent[27], a recommendation for NSAIDs dose adjustment and selection based on *CYP2C9* genotype was issued by CPIC[26]. To assess the impact of *CYP2C9* in NSAID-related ADEs, we analyzed the TPMI data to evaluate the association between *CYP2C9* activity and NSAID-related upper GI and renal events (Supplementary Table 10). We found that those with the *CYP2C9* PM phenotype were predisposed to NSAID-induced upper GI bleeding, although the sample size is relatively small (Fig. 6 and Supplementary Table 11). It was important to note that comorbidities (such as hypertension, diabetes, and cardiovascular disease) and concurrent drugs (e.g., diuretics, ACE inhibitors) contribute much more significantly to the adverse events of NSAIDs than *CYP2C9* status, as observed in Supplementary Table 10.

## Discussion

The benefits of pharmacogenetic testing in preventing severe adverse events are well documented[2]. Several randomized controlled trials and studies have highlighted the benefit of PGx-guided therapy, showcasing its potential in optimized drug selection and dosing that leads to improved efficacy and safety[28–31]. However, these small, mostly European studies do not address the applicability of PGx-guided therapy in non-European populations such as the Han Chinese, where literally

## a. *NUDT15* LOF

| Outcome | Tolerant(NM) | Event(NM) | Tolerant(LOF) | Event(LOF) | OR(95% CI) | P-value |
|---|---|---|---|---|---|---|
| ADE | 5,618 | 1,145 | 1,297 | 354 | 1.36(1.19-1.56) | 9.61E-06 |
| Leucopenia | 6,128 | 635 | 1,389 | 262 | 1.85(1.58-2.17) | 4.47E-14 |
| Thrombocytopenia | 6,316 | 447 | 1,525 | 126 | 1.20(0.97-1.48) | 0.10 |
| Hepatitis | 6,450 | 313 | 1,576 | 75 | 0.99(0.76-1.29) | 0.92 |
| Allergy | 6,724 | 39 | 1,640 | 11 | 1.06(0.53-2.13) | 0.87 |
| GI | 6,696 | 67 | 1,630 | 21 | 1.39(0.85-2.29) | 0.19 |
| HairLoss | 6,719 | 44 | 1,638 | 13 | 1.16(0.61-2.20) | 0.66 |

## b. *NUDT15* IM

| Outcome | Tolerant(NM) | Event(NM) | Tolerant(IM) | Event(IM) | OR(95% CI) | P-value |
|---|---|---|---|---|---|---|
| ADE | 5,618 | 1,145 | 1,260 | 322 | 1.28(1.11-1.48) | 5.44E-04 |
| Leucopenia | 6,128 | 635 | 1,346 | 236 | 1.73(1.46-2.04) | 9.44E-11 |
| Thrombocytopenia | 6,316 | 447 | 1,467 | 115 | 1.14(0.91-1.42) | 0.24 |
| Hepatitis | 6,450 | 313 | 1,512 | 70 | 0.95(0.73-1.26) | 0.74 |
| Allergy | 6,724 | 39 | 1,572 | 10 | 0.99(0.48-2.05) | 0.98 |
| GI | 6,696 | 67 | 1,563 | 19 | 1.32(0.79-2.21) | 0.30 |
| HairLoss | 6,719 | 44 | 1,574 | 8 | 0.80(0.37-1.71) | 0.56 |

## c. *NUDT15* PM

| Outcome | Tolerant(NM) | Event(NM) | Tolerant(PM) | Event(PM) | OR(95% CI) | P-value |
|---|---|---|---|---|---|---|
| ADE | 5,618 | 1,145 | 37 | 32 | 4.05(2.47-6.64) | 2.88E-08 |
| Leucopenia | 6,128 | 635 | 43 | 26 | 5.89(3.51-9.90) | 1.96E-11 |
| Thrombocytopenia | 6,316 | 447 | 58 | 11 | 2.67(1.35-5.30) | 4.91E-03 |
| Hepatitis | 6,450 | 313 | 64 | 5 | 1.76(0.70-4.41) | 0.23 |
| Allergy | 6,724 | 39 | 68 | 1 | 2.75(0.37-20.38) | 0.32 |
| GI | 6,696 | 67 | 67 | 2 | 3.17(0.75-13.28) | 0.12 |
| HairLoss | 6,719 | 44 | 64 | 5 | 10.84(3.72-31.58) | 1.24E-05 |

## d. *TPMT* LOF

| Outcome | Tolerant(NM) | Event(NM) | Tolerant(LOF) | Event(LOF) | OR(95% CI) | P-value |
|---|---|---|---|---|---|---|
| ADE | 6,745 | 1,463 | 201 | 40 | 0.92(0.65-1.31) | 0.64 |
| Leucopenia | 7,335 | 873 | 214 | 27 | 1.03(0.68-1.57) | 0.89 |
| Thrombocytopenia | 7,650 | 558 | 226 | 15 | 0.97(0.57-1.66) | 0.93 |
| Hepatitis | 7,829 | 379 | 231 | 10 | 0.86(0.44-1.69) | 0.67 |
| Allergy | 8,159 | 49 | 240 | 1 | 0.70(0.10-5.06) | 0.72 |
| GI | 8,122 | 86 | 239 | 2 | 0.84(0.20-3.43) | 0.81 |
| HairLoss | 8,154 | 54 | 238 | 3 | 1.28(0.31-5.28) | 0.74 |

## e. *TPMT* IM

| Outcome | Tolerant(NM) | Event(NM) | Tolerant(IM) | Event(IM) | OR(95% CI) | P-value |
|---|---|---|---|---|---|---|
| ADE | 6,745 | 1,463 | 200 | 40 | 0.92(0.65-1.31) | 0.66 |
| Leucopenia | 7,335 | 873 | 213 | 27 | 1.04(0.68-1.58) | 0.87 |
| Thrombocytopenia | 7,650 | 558 | 225 | 15 | 0.98(0.57-1.67) | 0.93 |
| Hepatitis | 7,829 | 379 | 230 | 10 | 0.87(0.44-1.70) | 0.68 |
| Allergy | 8,159 | 49 | 239 | 1 | 0.70(0.10-5.08) | 0.72 |
| GI | 8,122 | 86 | 238 | 2 | 0.84(0.21-3.45) | 0.81 |
| HairLoss | 8,154 | 54 | 237 | 3 | 1.28(0.31-5.32) | 0.73 |

**Fig. 4 | Influence of *NUDT15* and *TPMT* for AZA discontinuation due to ADE.** Forest plot of the ADE risk in AZA users with different *NUDT15* and *TPMT* phenotypes: (**a**) people who carried at least one *NUDT15* LoF alleles vs non-carriers, (**b**) *NUDT15* IM vs NM, (**c**) *NUDT15* PM vs NM, (**d**) people who carried at least one *TPMT* LoF alleles vs non-carriers, (**e**) *TPMT* IM vs NM. The case number, OR, 95% CI, and *P* value were listed in the table. The ORs and 95% CIs were estimated using logistic regression adjusting for covariates (two-sided test). Data points represent odds ratios; error bars represent 95% confidence intervals. Significant associations are shown in red. ADE = adverse events; CI = confident interval; GI = GI discomfort; IM = intermediate metabolizer; LoF = loss-of-function; NM = normal metabolizer; OR = odds ratio; PM = poor metabolizer.

everyone has PGx risk variants that affect his/her response to drugs developed with clinical trials conducted mostly with subjects of European ancestry. Well-recognized differences in drug response across populations, driven by genetic and clinical factors, have resulted in distinct optimal dosing recommendations for Asians and Europeans in current clinical practice. For instance, the US FDA notes a higher risk of myopathy in patients with decreased or poor function of *SLCO1B1* when taking 80 mg of simvastatin[6], and CPIC recommends a daily dosage of simvastatin below 20 mg for patients with decreased function of *SLCO1B1*[21]. However, the post prescribed and effective dose of simvastatin in Taiwan ranges from 10-20mg[32–34]. These observations emphasize the need for expanding PGx studies in underrepresented populations to refine guidelines and improve precision medicine globally. Our findings, along with previous studies, confirm the prevalence of clinically actionable PGx variants in Asian populations. For instance, studies on Sri Lankan, Indian, South Korean, Thai, and

## a. Atorvastatin vs *ABCG2* poor function phenotype

| Outcome | Tolerant(normal) | Event(normal) | Tolerant(poor) | Event(poor) | OR(95% CI) | P-value |
|---|---|---|---|---|---|---|
| SAM | 33,365 | 415 | 7,416 | 110 | 1.20(0.97-1.49) | 0.09 |
| sSAM | 33,735 | 45 | 7,507 | 19 | 1.92(1.12-3.29) | 0.02 |
| Myalgia | 33,365 | 368 | 7,416 | 91 | 1.12(0.89-1.42) | 0.32 |
| Myositis | 33,365 | 38 | 7,416 | 19 | 2.27(1.31-3.94) | 3.58E-03 |

## b. Fluvastatin vs *ABCG2* decreased function phenotype

| Outcome | Tolerant(normal) | Event(normal) | Tolerant(decreased) | Event(decreased) | OR(95% CI) | P-value |
|---|---|---|---|---|---|---|
| SAM | 1,964 | 40 | 1,739 | 53 | 1.50(0.99-2.27) | 0.06 |
| sSAM | 1,991 | 13 | 1,787 | 5 | 0.43(0.15-1.20) | 0.11 |
| Myalgia | 1,964 | 27 | 1,739 | 47 | 1.97(1.22-3.18) | 5.39E-03 |
| Myositis | 1,964 | 12 | 1,739 | 4 | 0.38(0.12-1.17) | 0.09 |

## c. Simvastatin vs *ABCG2* poor function phenotype

| Outcome | Tolerant(normal) | Event(normal) | Tolerant(poor) | Event(poor) | OR(95% CI) | P-value |
|---|---|---|---|---|---|---|
| SAM | 5,298 | 83 | 1,127 | 23 | 1.34(0.84-2.13) | 0.22 |
| sSAM | 5,369 | 12 | 1,141 | 9 | 3.69(1.55-8.80) | 3.17E-03 |
| Myalgia | 5,296 | 70 | 1,125 | 15 | 1.03(0.59-1.81) | 0.91 |
| Myositis | 5,296 | 10 | 1,125 | 7 | 3.43(1.30-9.05) | 0.01 |
| Rhabdomyolysis | 5,296 | 2 | 1,125 | 2 | 5.03(0.71-35.77) | 0.11 |

## d. Atorvastatin vs *SLCO1B1* decreased or poor function phenotype

| Outcome | Tolerant(normal) | Event(normal) | Tolerant(LOF) | Event(LOF) | OR(95% CI) | P-value |
|---|---|---|---|---|---|---|
| SAM | 56,994 | 711 | 15,115 | 238 | 1.27(1.09-1.47) | 1.72E-03 |
| sSAM | 57,621 | 84 | 15,325 | 28 | 1.26(0.82-1.93) | 0.30 |
| Myalgia | 56,994 | 625 | 15,115 | 209 | 1.27(1.08-1.48) | 3.35E-03 |
| Myositis | 56,994 | 67 | 15,115 | 23 | 1.30(0.81-2.09) | 0.28 |
| Rhabdomyolysis | 56,994 | 17 | 15,115 | 5 | 1.10(0.41-2.99) | 0.85 |

## e. Simvastatin vs *SLCO1B1* decreased or poor function phenotype

| Outcome | Tolerant(normal) | Event(normal) | Tolerant(LOF) | Event(LOF) | OR(95% CI) | P-value |
|---|---|---|---|---|---|---|
| SAM | 8,868 | 132 | 2,504 | 53 | 1.42(1.03-1.96) | 0.03 |
| sSAM | 8,983 | 17 | 2,543 | 14 | 2.91(1.43-5.91) | 3.14E-03 |
| Myalgia | 8,865 | 114 | 2,503 | 40 | 1.24(0.86-1.79) | 0.24 |
| Myositis | 8,865 | 16 | 2,503 | 9 | 1.99(0.88-4.52) | 0.10 |
| Rhabdomyolysis | 8,865 | 1 | 2,503 | 5 | 17.72(2.07-151.75) | 8.70E-03 |

## f. Fluvastatin vs *CYP2C9* LOF

| Outcome | Tolerant(NM) | Event(NM) | Tolerant(LOF) | Event(LOF) | OR(95% CI) | P-value |
|---|---|---|---|---|---|---|
| SAM | 3,809 | 94 | 304 | 7 | 0.93(0.43-2.03) | 0.86 |
| sSAM | 3,884 | 19 | 309 | 2 | 1.32(0.31-5.71) | 0.71 |
| Myalgia | 3,809 | 75 | 304 | 4 | 0.67(0.24-1.84) | 0.44 |
| Myositis | 3,809 | 17 | 304 | 2 | 1.47(0.34-6.40) | 0.61 |

**Fig. 5 | Influence of *ABCG2*/*SLCO1B1*/*CYP2C9* in SAMs.** Forest plot of the SAM risk in statin users with different *ABCG2*, *SLCO1B1*, or *CYP2C9* phenotypes: (**a**) atorvastatin users who carried two *ABCG2* LoF alleles vs non-carriers, (**b**) fluvastatin users who carried one *ABCG2* LoF alleles vs non-carriers, (**c**) simvastatin users who carried two *ABCG2* LoF alleles vs non-carriers, (**d**) atorvastatin users who carried at least one *SLCO1B1* LoF alleles vs non-carriers, (**e**) simvastatin users who carried at least one *SLCO1B1* LoF alleles vs non-carriers, and (**f**) fluvastatin users who carried at least one *CYP2C9* LoF alleles vs non-carriers. The case number, OR, 95% CI, and *P* value were listed in the table. The ORs and 95% CIs were estimated using logistic regression adjusting for covariates (two-sided test). Data points represent odds ratios; error bars represent 95% confidence intervals. Significant associations are shown in red. CI = confident interval; IM = intermediate metabolizer; LoF = loss-of-function; NM = normal metabolizer; OR = odds ratio. SAM = statin-associated muscle events, including myalgia, myositis, and rhabdomyolysis; sSAM = severe forms of statin-associated muscle events, specifically myositis and rhabdomyolysis.

## a. *CYP2C9* LOF

| Outcome | Tolerant(NM) | Event(NM) | Tolerant(LOF) | Event(LOF) | OR(95% CI) | P-value |
|---------|--------------|-----------|---------------|------------|------------|---------|
| ADE | 8,757 | 499 | 705 | 30 | 0.75(0.51-1.10) | 0.14 |
| GI | 9,007 | 249 | 719 | 16 | 0.80(0.48-1.34) | 0.40 |
| kidney | 8,757 | 499 | 706 | 29 | 0.72(0.49-1.07) | 0.10 |

## b. *CYP2C9* IM

| Outcome | Tolerant(NM) | Event(NM) | Tolerant(IM) | Event(IM) | OR(95% CI) | P-value |
|---------|--------------|-----------|--------------|-----------|------------|---------|
| ADE | 8,757 | 499 | 693 | 28 | 0.71(0.47-1.05) | 0.09 |
| GI | 9,007 | 249 | 707 | 14 | 0.71(0.41-1.23) | 0.23 |
| kidney | 8,757 | 499 | 693 | 28 | 0.71(0.47-1.05) | 0.09 |

## c. *CYP2C9* PM

| Outcome | Tolerant(NM) | Event(NM) | Tolerant(PM) | Event(PM) | OR(95% CI) | P-value |
|---------|--------------|-----------|--------------|-----------|------------|---------|
| ADE | 8,757 | 499 | 12 | 2 | 3.37(0.75-15.14) | 0.11 |
| GI | 9,007 | 249 | 12 | 2 | 5.97(1.33-26.87) | 0.02 |
| kidney | 8,757 | 499 | 13 | 1 | 1.56(0.20-11.93) | 0.67 |

**Fig. 6 | Influence of *CYP2C9* in NSAID-associated adverse events.** Forest plot of the ADE risk in statin users with different *CYP2C9* phenotypes: (**a**) people who carried at least one *CYP2C9* LoF alleles vs non-carriers, (**b**) *CYP2C9* IM vs NM, and (**c**) *CYP2C9* PM vs NM. The case number, OR, 95% CI, and *P* value were listed in the table. The ORs and 95% CIs were estimated using logistic regression adjusting for covariates (two-sided test). Data points represent odds ratios; error bars represent 95% confidence intervals. Significant associations are shown in red. CI = confident interval; GI = GI discomfort; IM = intermediate metabolizer; LoF = loss-of-function; NM = normal metabolizer; OR = odds ratio; PM = poor metabolizer.

Chinese populations highlight significant differences in the frequency of PGx variants compared to Europeans. Variants like *NUDT15*, associated with AZA-induced myelotoxicity, and *CYP2C19*, influencing clopidogrel metabolism, are notably more prevalent in Asians[35–39].

Our retrospective study of the 4 gene-drug pairs in 486,956 participants of the TPMI, for whom we have both genetic and longitudinal clinical data, shows that PGx-guided therapy is not straightforward. First, while our findings validate the published results that PGx risk variants increase the risk of adverse events, and the increase is statistically significant, the relative risk is low or moderate. Second, the vast majority of those with PGx risk variants who take the drug in question do not suffer from the predicted side-effects. Conversely, a significant fraction of those without PGx risk variants suffer from adverse events. This underscores that PGx is not an absolute predictor of ADEs but a critical tool that complements other clinical factors, such as patient history, comorbidities, and drug-drug interactions, to guide treatment decisions. When used in conjunction with these factors, PGx information can help identify at-risk patients and tailor therapies more effectively. Third, many of the adverse events are reversible, non-life-threatening events, and can be managed easily. For example, statin-associated myalgia, characterized by muscle pain or weakness, occurs in up to 15% of treated patients. Most cases are mild and completely resolved upon discontinuation of the statin, with symptoms improving within an average of 2–3 months. In rare cases, rechallenging with a different statin may result in successful tolerance without recurrent symptoms[25,40,41]. Finally, as there are no equally effective drug alternatives available in many cases, avoiding adverse events by not taking a drug means that one is taking a drug of lower efficacy. For instance, clopidogrel is a widely used P2Y12 inhibitor in elderly patients with ACS or those undergoing percutaneous intervention (PCI). While alternatives like ticagrelor or prasugrel offer enhanced antiplatelet effects, they are associated with significantly higher risks of bleeding, making clopidogrel the safer and more practical option for patients older than age 70 or with higher bleeding risk[42–44]. This underscores the clinical challenge of balancing efficacy, safety, and patient-specific factors when managing pharmacogenetic risks.

Given our findings, one must proceed with caution when implementing PGx-guided therapy. Existing resources, such as the clinical practice recommendations from CPIC and US FDA, already provide valuable recommendations for risk mitigation and management strategies in some scenarios. However, further studies must be conducted to (1) identify additional (genetic and non-genetic) factors that cause ADEs that explain the baseline occurrence of such events in those without PGx risk variants; (2) explore protective factors that allow carriers of PGx risk variants to tolerate drugs without adverse events; and (3) develop and refine risk-management strategies for PGx risk variant carriers, particularly for scenarios where CPIC and US FDA have not yet offer comprehensive recommendations. Such efforts will ensure that individuals can safely benefit from the most effective therapies while minimizing risks.

Although our findings and conclusions are strong for the 4 gene-drug pairs studied, and the results can be extrapolated to other drugs, our study has several limitations. First, the TPMI obtains clinical data for each participant only from the hospital where they are enrolled. In Taiwan, due to the convenience of the National Health Insurance system and the close proximity of hospitals, patients often receive care from multiple hospitals. For instance, a patient might visit one hospital for diabetes management and another for renal disease. However, the TPMI dataset does not capture clinical data from other hospitals, which could lead to under-reporting and incomplete datasets. Second, some drugs, such as NSAIDs, may be available over the counter (OTC) in Taiwan. Since the TPMI dataset is based on prescription records obtained from hospitals, OTC drug use is not captured. This could lead to underestimation of the use for drugs that are commonly obtained without a prescription, potentially impacting the assessment of drug-related outcomes. Third, the genetic data are based on SNP-array data,

which means that not all the PGx risk variants in each pharmacogene are represented. For example, some variants are embedded in repeat regions where no suitable probes can be designed on the array. In addition, variants of extremely low allele frequency (<0.1% minor allele frequency) cannot be typed accurately on the array and are therefore excluded from the design. Fourth, the clinical phenotypes are extracted from the clinical data based on chart-review and the variability in clinical note style and substance across hundreds of doctors from 33 hospitals create noise in the data. Some cases and controls are likely excluded due to this reason. Apart from genetic variants, the heterogeneity in drug response may also stem from other factors, including environmental and nutritional influences, disease severity, comorbidities, concomitant drugs, and individual patient lifestyles. To overcome these limitations, future studies will be conducted with comprehensive clinical data from the national health insurance database (that collects information from all hospitals and clinics the participants obtain their care), comprehensive genetic data from whole genome sequencing (WGS), and prospective study with standardized recording of clinical outcomes.

Phenoconversion, the alteration of an individual's observed drug-response phenotype due to external or environmental factors such as concurrent drugs, comorbidities, or lifestyle, is another important consideration[45–47]. Although we adjusted for comorbidities and concurrent drugs, phenoconversion was not explicitly evaluated. This limitation could potentially confound the observed associations between PGx variants and clinical outcomes. For example, concurrent drugs that inhibit or induce drug-metabolizing enzymes could mask or amplify the effects of PGx variants, leading to misclassification of phenotypes. Future studies should incorporate phenoconversion explicitly by leveraging detailed drug histories and environmental data to better distinguish genetic from non-genetic influences on drug response. Addressing phenoconversion will help refine genotype-phenotype associations and improve the clinical utility of PGx-guided therapy. Finally, the current study does not include analysis of the risk variants in *CYP2D6*, the gene that encodes a pivotal enzyme in the metabolic pathways of nearly 20% of frequently prescribed drugs[48]. The challenge of accurately typing *CYP2D6* is due to the high degree of polymorphism and complex nature with structural variations (SV) in the gene[49]. The lack of multiplex PCR and SV analysis algorithm make accurately determining *CYP2D6* status from SNP array data impossible at this time. An optimal pipeline to overcome this challenge is under construction.

In conclusion, we report results from the largest retrospective study in non-Europeans on the distribution of PG risk variants and their impact for 4 widely accepted gene-drug pairs on drug-related adverse events and treatment responses. Our findings show that implementing PGx-guided therapy in large populations is not simple and should be done with caution. We speculate that these conclusions apply to many other gene-drug pairs and to non-European populations, highlighting the need for comprehensive, integrative strategies to enhance the safety and efficacy of precision medicine.

## Methods
### Data source
This study was conducted with genetic and clinical datasets from the TPMI cohort, with participants recruited from 16 medical centers (encompassing 33 hospitals) in Taiwan. Genetic data includes the whole genome genotyping data on custom-designed TPMI SNP arrays, TPM and TPM2 array, genome-wide imputation data, and *HLA* imputation data. TPM (686,463 SNPs) and TPM2 (743,227 SNPs) arrays are designed by TPMI and Thermo Fisher Scientific for the TPMI project, specifically for the Han Chinese population with superior coverage for GWAS-grid and previously published health related variants. The EMRs were provided by the hospital through which the participants were enrolled. They included outpatient, inpatient, and emergency room

visiting records, drug prescription records, discharge summaries and operation notes, laboratory test results, and reports of pathology, surgery, imaging, and Mini-Mental State Examination. The 1498 WGS data from the Taiwan Biobank (TWB) were used for the imputation reference panel and for genotype validation[50]. The DeepVariant variant calling pipelines were used, and SHAPEIT5 and IMPUTE5 were applied for phasing and imputation[51–53]. For the imputation of *G6PD*, which is located on chromosome X, we utilized a reference panel derived from samples of TPM2 due to the absence of relevant data points in TPM1. The imputation process was conducted using Shapeit4 and Impute5. Male samples were imputed with the −haploid parameter in Impute5 to account for the haploid nature of the X chromosome in males, while female samples underwent standard diploid imputation. The training and validating dataset for Hibag encompassed 1359 HLA typing data sourced from multiple repositories, including the Adverse Drug Reaction Project, the Collaborative Study to Establish a Cell Bank and a Genetic Database on Non-Aboriginal Taiwanese, and the TWB. Utilizing the Hibag algorithm, classical *HLA* alleles such as *HLA-A*, *HLA-B*, *HLA-C*, *HLA-DRB1*, *HLA-DQB1*, and *HLA-DPB1* were imputed. Each *HLA* allele underwent training with a 500-kb flanking region and utilized 500 classifiers to construct models, ensuring comprehensive coverage and robust predictive accuracy[50]. This study was approved by TPMI's committees, and the data usage followed the approval from the ethical committees of the Academia Sinica (AS-IRB01-18079), the TWB (TWBR10806-04), and all participating hospitals: Taipei Veterans General Hospital (2020-08-014 A), National Taiwan University Hospital (201912110RINC), Tri-Service General Hospital (2-108-05-038), Chang Gung Memorial Hospital (201901731A3), Taipei Medical University Healthcare System (N202001037), Chung Shan Medical University Hospital (CS19035), Taichung Veterans General Hospital (SF19153A), Changhua Christian Hospital (190713), Kaohsiung Medical University Chung-Ho Memorial Hospital (KMUHIRB-SV(II)−20190059), Hualien Tzu Chi Hospital (IRB108-123-A), Far Eastern Memorial Hospital (110073-F), Ditmanson Medical Foundation Chia-Yi Christian Hospital (IRB2021128), Taipei City Hospital (TCHIRB-10912016), Koo Foundation Sun Yat-Sen Cancer Center (20190823 A), Cathay General Hospital (CGH-P110041), andFu Jen Catholic University Hospital (FJUH109001).

### Pharmacogenomic variants analysis
The pharmacogenomic variants analyzed in this study were primarily obtained through genotyping performed using custom-designed TPMI SNP arrays, TPMv1 (686,463 SNPs) and TPMv2 (743,227 SNPs). These arrays were tailored for the Han Chinese population to optimize coverage for GWAS studies and pharmacogenetically relevant markers. Imputation was then performed using the TWB WGS dataset as the reference panel. This process utilized advanced phasing and imputation tools, SHAPEIT5 and IMPUTE5, ensuring robust data quality and coverage. The drug-gene pairs were adopted from US FDA Table of Pharmacogenetic Associations[6] and the Clinical Guideline Annotations table of CPIC guidelines from PharmGkB[54] (accessed on 2024/02/27). The drugs with fewer than 10 prescription records were removed from the analysis. The curation of PGX variants and actionable PGx phenotype were based on the instructions from PharmGkB[55] and PharmVar[56]. The PGx variants were first screened with WGS data to remove the ones with allele frequency lower than 0.1%. To further ensure the accuracy of key actionable PGx variants, we conducted a validation process on a representative subset of samples. This targeted validation utilized multiple technologies, including WGS, Sanger sequencing, and the Sequenom MassARRAY platform. This step was not performed for the entire cohort of 486,956 participants but focused on confirming the reliability of genotype calls and imputed variants for clinically actionable alleles. The variants with sensitivity and specificity higher than 99% were saved for further analysis. This multi-layered approach of genotyping, imputation, and targeted validation was designed to enhance data reliability while balancing the scale of the study. The final

list of drug-gene pairs evaluated in this study is found in Supplemental Table 2, and the variants information in Supplemental Table 1.

### Drug prescription and treatment outcome assessment

When analyzing the number of individuals encountered drugs with PGx warnings, 58 approved drugs with clear genetic-base clinical utility recommendations were selected based on their availability in the TPMI dataset. Individuals with only a single prescription record for each drug were excluded.

The treatment outcome assessment focused on 4 drug-gene pairs, azathioprine (NUDT15/TPMT), clopidogrel (CYP2C19), statins (ABCG2/CYP2C9/SLCO1B1), and NSAIDs (CYP2C9)[6,21,57–59]. Data from people taking AZA, clopidogrel, statins, and NSAID metabolized by CYP2C9 were analyzed to assess the influence of PGx variants and phenotypes on drug response. To assess the impact of pharmacogenomic (PGx) variants on drug response, we analyzed prescription records and treatment outcomes for four gene-drug pairs: azathioprine (NUDT15/TPMT), clopidogrel (CYP2C19), statins (ABCG2/CYP2C9/SLCO1B1), and NSAIDs (CYP2C9). Prescription records were examined to define the duration of drug exposure, and ADEs were identified by reviewing corresponding time intervals following drug discontinuation. Both structured and unstructured EMRs were utilized, including hospitalization records, surgical reports, imaging findings, laboratory results, and physician notes. Unstructured physician notes provided valuable insights into ADEs and reasons for drug discontinuation. Collaborating physicians and pharmacists reviewed these records to validate reported outcomes. Data cleaning and standardization processes were conducted by the TTPMI team, integrating clinical test results and structured data, such as ICD codes and laboratory values, to reduce variability across participating institutions.

### Clopidogrel and CYP2C19

To study the influence of CYP2C19 on clopidogrel response, we included clopidogrel users aged ≥18 years who demonstrated good compliance and lacked severe allergic reactions. The primary endpoint was MACE, including CV death, non-fatal HF, non-fatal UA, acute MI, acute ischemic stroke or transient ischemic attack, or TLR requiring clinical interventions such as PCI or surgical bypass (extracted from image and operation reports). Compliance was assessed by reviewing descriptions extracted from outpatient medical records. Since the TPMI medical records contain unstructured physician notes, collaborating physicians and pharmacists reviewed these records to identify information regarding compliance. When clarification was needed, we consulted with the physicians to confirm the content.

### Azathioprine and NUDT15/TPMT

The AZA study cohort comprised of AZA tolerant controls and patients who discontinued AZA due to massive hair loss, GI discomfort (nausea, vomiting, and diarrhea), allergic reactions, hepatitis (defined as ALS/AST > 3x ULN), leucopenia (defined as WBC < 3500/mm$^3$), and thrombocytopenia (defined as platelet count <150,000 /uL), as documented by the responsible physicians and/or laboratory test results. Allergic reactions were defined as hypersensitivity events documented by the responsible physicians in the medical records, including symptoms such as rash, urticaria, or other similar reactions attributed to AZA use. Patients with pre-existing hematological malignancies or poor liver function were excluded. The study design is shown in Supplementary Fig. 2.

### Statins and ABCG2/CYP2C9/SLCO1B1

The TPMI participants taking atorvastatin, fluvastatin, lovastatin, pitavastatin, pravastatin, rosuvastatin, or simvastatin without any muscular disorder history were included to assess the role of ABCG2, CYP2C9, and SLCO1B1 in the risk of SAMs (CPK elevation with muscle complaints, myalgia, myositis, and rhabdomyolysis).

### NSAIDs and CYP2C9

For NSAID users, we assessed adverse renal events and upper GI discomfort in different CYP2C9 phenotype groups if they had prescriptions of NSAIDs mainly metabolized via CYP2C9 (celecoxib, flurbiprofen, ibuprofen, meloxicam, piroxicam, and tenoxicam) but without history of end stage kidney disease (ESRD, ICD10 code: N18.5, and N19), poor renal function (<15 ml/min/1.73 m$^2$), receiving dialysis (NHIRD order code: 58001 C, 58002 C, 58009B, 58010B, 58011 C, 58012B, 58013 C, 58017 C, 58018 C, and 58026 C), renal replacement therapy (NHIRD order code: 76020B and N26028 and ICD10 code: T86.1 and Z94.0), alcoholism (ICD10 code: F10), esophageal varices (ICD10 code: I85 and I98.2), Mallory–Weiss syndrome (ICD10 code: K22.6), liver cirrhosis (ICD10 code: K70, K72–74, and K76), GI tract cancer (ICD10 code: C15, C16, and C17), and coagulation defects (ICD10 code: D65-D68). Comorbidities included diabetes mellitus (ICD10 code: E08-E13), hypercholesterolemia (ICD10 code: E78.0-E78.5), hypertension (I10), ischemic heart diseases (ICD10 code: I20-I25), HF (ICD10 code: I50), and cerebrovascular diseases (ICD10 code: I60-I69) were analyzed as well to identify their influence on treatment outcome[60,61].

### Statistical analysis

The frequencies of PGx variants from genotype and WGS data were calculated by PLINK and VCFtools. Fisher exact test and chi-square tests were conducted to compare differences between tolerant and non-tolerant individuals who carry the different PGx variants. Continuous variables were shown as means along with their standard deviations, whereas categorical variables were displayed as numerical counts and percentages. For quantitative data, comparison between 2 groups was performed by 2-tailed student t-test. Logistic regression analysis was utilized to determine the odds ratio (OR) and 95% confidence interval of the allele model. The multivariate logistic regression was adjusted for clinical factors, including sex, age, comorbidities, and concurrent drugs. A P value ≤ 0.05 was considered significant in this study. Statistical analyses were conducted using Python (version 3.11).

## Data availability

All summary statistics and results from this study are freely available from the TPMI website (https://tpmi.ibms.sinica.edu.tw). In compliance with the privacy laws governing genetic and health data in Taiwan, the de-identified TPMI clinical and genotyping data are kept in a secure server at the Academia Sinica and not released to the public. TPMI is in the process of transitioning to a biobank model, which will make the TPMI data openly accessible to the public for research purposes in about 18 months. Researchers who wish to access the individual clinical and genotyping data prior to that time are welcome to submit a simple request through the TPMI website. The TPMI Data Access Committee will respond within 2 weeks of the submission and facilitate the process for data access. In addition to the TPMI data, we analyzed the TWB dataset as part of the validation, and the genotype data from the TWB are available through a formal application process (https://www.twbiobank.org.tw/index.php). Source data are provided with this paper.

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

## Acknowledgements

We thank all the participants and researchers of the Taiwan Precision Medicine Initiative and the Taiwan Biobank. This study was funded in part by the Academia Sinica (40-05-GMM, AS-GC-110-MD02, and 236e-1100202 to P.-Y.K. and J.-Y.W.) and the National Development Fund, Executive Yuan (NSTC 111-3114-Y-001-001 to P.-Y.K.).

## Author contributions

C.Y.W., P.Y.K., M.S.W., C.K.C., Y.J.S., Yuan-Tsong C., and J.Y.W. were involved in the conceptualization and design of the study. M.S.L., S.P.W., K.T.L., H.P.C., Ying-Ju C., J.S., Ying-Ting C., Chiung-Chih C., C.F.K., J.C.L., H.C.K., T.M.C., C.W.L., J.H.L., S.F.L., H.T.C., L.Y.L., L.C.C., C.T.T., H.L.K., J.J.Y., J.S.J., M.C.C., T.C.H., S.F.Y., H.J.L., S.C.S., P.L.C., P.F.L., C.L.T., C.K.T., S.E.T., C.M.L., Yung-Fu W., Chih-Yang H., S.Z.L., Chun-Chun C., T.K.L., S.M.H., Chih-Hung C., C.D.C., G.C.M., T.Y.C., J.J.H., Chien-Lin L., K.J.K., Chen-Fang H., S.S.C., P.Y.C., K.C.T., Y.C.P., S.J.W., S.P.C., M.T.T., T.I.H., C.W.S., D.C.T., C.C.H., J.L.F., K.H.L., Y.T.L., Ching-Liang L., Y.C.L., Y.H.H., C.P.L., Yen-Feng W., Y.C.H., Y.M.C., T.H.H., Ching-Heng L., Yen-Ju C., I.C.C., C.L.M., S.J.C., Yen-Lin C., Y.J.L., Chih-Hung L., W.J.L., H.T., T.T.Y., Hsin-Chien Y., J.H.C., Chun-Yao H., L.C., Y.W.L., B.Y.H., C.J.H., Y.K.L., Y.F.L., T.C.C., D.C.W., J.Y.K., C.Y.H., S.C.C., C.C.L., Chung-Feng H., C.C.S., L.J.Y., and Chung-Hwan C. conducted the investigation. M.F.T., E.C.Y., K.M.C., S.M.C., M.S.L., S.P.W., K.T.L., and Ying-Ting C. curated the data. C.Y.W., M.F.T., E.C.Y., K.M.C., and S.M.C. performed the formal analysis. T.C.Y., S.L.L., J.Y.W., L.H.L., Chun-houh C., C.S.-J.F., Hsin-Chou Y., Y.T.H., Y.M.L., Chien-Hsiun C., Chih-Cheng C., H.S.C., Yen-Ling C., H.C.C., and P.Y.K. supervised the project. C.Y.W. and P.Y.K. wrote the initial draft. C.Y.W., J.Y.W., M.F.T., L.H.L., Chun-houh C., C.S.-J.F., H.C.Y., Y.T.H., H.H.C., Y.M.L., E.C.Y., Chien-Hsiun C., and P.Y.K. reviewed and edited the manuscript, and all authors were involved in the revision of the manuscript for publication. J.Y.W., Chih-Cheng C., H.S.C., Yen-Ling C., and H.C.C. were involved in resources. Y.M.L., H.P.C., Ying-Ju C., J.S., and P.Y.K. were involved in project administration. J.Y.W., Y.M.L., and P.Y.K. were involved in funding acquisition.

## Competing interests

The authors declare no competing interests.

## Additional information

Chun-Yu Wei [1,2,145] ✉, Ming-Shien Wen [3,4,145], Chih-Kuang Cheng[5,145], Yi-Jing Sheen[6,7,8,145], Tsung-Chieh Yao[9,145], Sing-Lian Lee[10,145], Jer-Yuarn Wu[1], Ming-Fang Tsai[1], Ling-Hui Li[1], Chun-houh Chen [11], Cathy S.-J. Fann [1], Hsin-Chou Yang [11,12], Yen-Tsung Huang [11,13,14], Hung-Hsin Chen [1], Yi-Min Liu[1], Erh-Chan Yeh [1], Yu-Ching Peng[15], Shuu-Jiun Wang [16,17,18], Shih-Pin Chen [16,19,20], Ming-Tsun Tsai[20,21], Teh-Ia Huo[22,23,24], Chien-Wei Su[20,25,26], Der-Cherng Tarng[21,26,27], Chin-Chou Huang[26,28], Jong-Ling Fuh [16,29], Keng-Hsin Lan[23,24,30], Yo-Tsen Liu [24,31,32], Ching-Liang Lu[26,33], Yi-Chung Lee [16,18,34], Yi-Hsiang Huang [30,35,36], Chung-Pin Li [37,38,39], Yen-Feng Wang[16,18,24], Yu-Cheng Hsieh[8,20,40], Yi-Ming Chen[8,40], Tzu-Hung Hsiao[40], Ching-Heng Lin[40], Yen-Ju Chen[40], I-Chieh Chen [40], Chien-Lin Mao[40], Shu-Jung Chang[40], Yen-Lin Chang [41], Yi-Ju Liao[41], Chih-Hung Lai [42], Wei-Ju Lee[8,43], Hsin Tung[8,43], Ting-Ting Yen[44], Hsin-Chien Yen[45], Jer-Hwa Chang [46,47], Chun-Yao Huang[48,49,50], Lung Chan [51,52,53], Yung-Wei Lin[54,55,56], Bu-Yuan Hsiao[48,49,50], Chaur-Jong Hu[51,52], Yung-Kuo Lin[48,57], Yung-Feng Lin[58], Tung-Cheng Chang[59,60], Deng-Chyang Wu[61,62,63], Jung-Yu Kan[64,65,66], Chung-Yao Hsu[67,68], Szu-Chia Chen[69,70,71], Ching-Chia Li[72,73], Chung-Feng Huang[74,75], Chau-Chyun Sheu [76,77], Lii-Jia Yang[69,78], Chung-Hwan Chen [79,80,81], Kuan-Mao Chen [1], Shu-Min Chang[1], Min-Shiuan Liou[1], Shi-Ping Wang[1], Kuan-Ting Lin [1], Hui-Ping Chuang[1], Ying-Ju Chen[1], Joey Sin [1], Ying-Ting Chen[1], Chiung-Chih Chang[82,83,84], Chang-Fu Kuo [85,86,87], Jing-Chi Lin[88], Ho-Chang Kuo [89,90], Tien-Min Chan [86,87], Chao-Wei Lee[91], Jenn-Haung Lai[86], Shue-Fen Luo[92], Hao-Tsai Cheng[93,94,95], Lian-Yu Lin[96,97,98], Li-Chun Chang[96,97], Chia-Ti Tsai[96,97], Hsien-Li Kao[96,97], Jian-Jyun Yu[98], Jiann-Shing Jeng[99,100], Min-Chin Chiu[98], Tzu-Chan Hong[97,101,102], Shun-Fa Yang [103,104], Hsueh-Ju Lu[105,106], Sheng-Chiang Su[107], Pauling Chu[108], Peng-Fei Li[107], Chia-Lin Tsai[109], Chia-Kuang Tsai[109], Shih-En Tang[110], Chien-Ming Lin[111], Yung-Fu Wu[112], Chih-Yang Huang [113,114], Shinn-Zong Ling[115,116], Chun-Chun Chang[117,118], Tzu-Kai Lin[119,120], Sheng-Mou Hsiao[121,122,123], Chih-Hung Chang[122,124], Chih-Dao Chen[125], Gwo-Chin Ma[126], Ting-Yu Chang[126], Juey-Jen Hwang[127,128], Chien-Lin Lu[128,129], Kuo-Jang Kao [130], Chen-Fang Hung[130], Shiou-Sheng Chen[131,132,133], Po-Yueh Chen[134,135], Kochung Tsui[136,137,138], Yuan-Tsong Chen[1], Chien-Hsiun Chen [1,146] ✉, Chih-Cheng Chien[136,139,146] ✉, Han-Sun Chiang[128,140,146] ✉, Yen-Ling Chiu [141,142,146] ✉, Hsiang-Cheng Chen [143,146] ✉ & Pui-Yan Kwok [1,144,146] ✉

[1]Institute of Biomedical Sciences, Academia Sinica, Taipei, Taiwan. [2]Core Laboratory of Neoantigen Analysis for Personalized Cancer Vaccine, Office of R&D, Taipei Medical University, Taipei, Taiwan. [3]Division of Cardiology, Chang Gung Memorial Hospital, Linkou Branch, Taoyuan, Taiwan. [4]School of Medicine, Chang Gung University, Taoyuan, Taiwan. [5]Department of Neurology, Chang Gung Memorial Hospital, Linkou Branch, Taoyuan, Taiwan. [6]Division of Endocrinology and Metabolism, Department of Internal Medicine, Taichung Veterans General Hospital, Taichung, Taiwan. [7]Department of Medicine, School of Medicine, National Yang Ming Chiao Tung University, Taipei, Taiwan. [8]Department of Post-Baccalaureate Medicine, College of Medicine, National Chung Hsing University, Taichung, Taiwan. [9]Division of Allergy, Asthma, and Rheumatology, Department of Pediatrics, Chang Gung Memorial Hospital, Taoyuan, Taiwan. [10]Department of Internal Medicine, Koo Foundation Sun Yat-Sen Cancer Center, Taipei, Taiwan. [11]Institute of Statistical Science, Academia Sinica, Taipei, Taiwan. [12]Biomedical Translation Research Center, Academia Sinica, Taipei, Taiwan. [13]Bioinformatics Program, Taiwan International Graduate Program, Academia Sinica, Taipei, Taiwan. [14]Department of Mathematics, Institute of Epidemiology and Preventive Medicine, National Taiwan University, Taipei, Taiwan. [15]Department of Pathology and Laboratory Medicine, Taipei Veterans General Hospital, Taipei, Taiwan. [16]Department of Neurology, Neurological Institute, Taipei Veterans General Hospital, Taipei, Taiwan. [17]College of Medicine, National Yang Ming Chiao Tung University, Taipei, Taiwan. [18]Brain Research Center, National Yang Ming Chiao Tung University, Taipei, Taiwan. [19]Division of Translational Research, Department of Medical Research, Taipei Veterans General Hospital, Taipei, Taiwan. [20]Institute of Clinical Medicine, National Yang Ming Chiao Tung University, Taipei, Taiwan. [21]Division of Nephrology, Department of Medicine, Taipei Veterans General Hospital, Taipei, Taiwan. [22]Division of Basic Research, Department of Medical Research, Taipei Veterans General Hospital, Taipei, Taiwan. [23]Institute of Pharmacology, College of Medicine, National Yang Ming Chiao Tung University, Taipei, Taiwan. [24]School of Medicine, National Yang Ming Chiao Tung University, Taipei, Taiwan. [25]Division of General Medicine, Department of Medicine, Taipei Veterans General Hospital, Taipei, Taiwan. [26]Faculty of Medicine, School of Medicine, National Yang Ming Chiao Tung University, Taipei, Taiwan. [27]Institute of Clinical Medicine, School of Medicine, National Yang Ming Chiao Tung University, Taipei, Taiwan. [28]Division of Cardiology, Department of Medicine, Taipei Veterans General Hospital, Taipei, Taiwan. [29]Brain Research Center & School of Medicine, National Yang Ming Chiao Tung University, Taipei, Taiwan. [30]Division of Gastroenterology and Hepatology, Department of Medicine, Taipei Veterans General Hospital, Taipei, Taiwan. [31]Division of Epilepsy, Neurological Institute, Taipei Veterans General Hospital, Taipei, Taiwan. [32]Institute of Brain Science, National Yang Ming Chiao Tung University, Taipei, Taiwan. [33]Division of Gastroenterology, Taipei Veterans General Hospital, Taipei, Taiwan. [34]Department of Neurology, National Yang Ming Chiao Tung University School of Medicine, Taipei, Taiwan. [35]Healthcare and Services Center, Taipei Veterans General Hospital, Taipei, Taiwan. [36]Institute of Clinical Medicine, College of Medicine, National Yang Ming Chiao Tung University, Taipei, Taiwan. [37]Division of Clinical Skills Training, Department of Medical Education, Taipei Veterans General Hospital, Taipei, Taiwan. [38]School of Medicine, College of Medicine, National Yang Ming Chiao Tung University, Taipei, Taiwan. [39]Therapeutic and Research Center of Pancreatic Cancer, Taipei Veterans General Hospital, Taipei, Taiwan. [40]Department of Medical Research, Taichung Veterans General Hospital, Taichung, Taiwan. [41]Department of Pharmacy, Taichung Veterans General Hospital, Taichung, Taiwan. [42]Department of Medicine and Cardiovascular Center, Taichung Veterans General Hospital, Taichung, Taiwan. [43]Neurological Institute, Taichung Veterans General Hospital, Taichung, Taiwan. [44]Department of Otolaryngology, Taichung Veterans General Hospital, Taichung, Taiwan. [45]Division of Pediatric Genetics and Metabolism, Children's Medical Center, Taichung Veterans General Hospital, Taichung, Taiwan. [46]School of Respiratory Therapy, College of Medicine, Taipei Medical University, Taipei, Taiwan. [47]Division of Pulmonary Medicine, Department of Internal Medicine, Wan Fang Hospital, Taipei Medical University, Taipei, Taiwan. [48]Division of Cardiology, Department of Internal Medicine, School of Medicine, College of Medicine, Taipei Medical University, Taipei, Taiwan. [49]Division of Cardiology and Cardiovascular Research Center, Department of Internal Medicine, Taipei Medical University Hospital, Taipei, Taiwan. [50]Taipei Heart Institute, Taipei Medical University, Taipei, Taiwan. [51]Department of Neurology, Shuang Ho Hospital, Taipei Medical University, New Taipei City, Taiwan. [52]Department of Neurology, School of Medicine, College of Medicine, Taipei Medical University, Taipei, Taiwan. [53]Taipei Neuroscience Institute, Shuang Ho Hospital, Taipei Medical University, New Taipei City, Taiwan. [54]Department of Urology, School of Medicine, College of Medicine and TMU Research Center of Urology and Kidney

(TMU-RCUK), Taipei Medical University, Taipei, Taiwan. [55]Department of Urology, Wan Fang Hospital, Taipei Medical University, Taipei, Taiwan. [56]International Master/PhD Program in Medicine, College of Medicine, Taipei Medical University, Taipei, Taiwan. [57]Division of Cardiovascular Medicine, Department of Internal Medicine, Wan Fang Hospital, Taipei Medical University, Taipei, Taiwan. [58]Department of Medicine Research, Taipei Medical University Hospital, Taipei, Taiwan. [59]Division of Colorectal Surgery, Department of Surgery, Shuang Ho Hospital, Taipei Medical University, New Taipei City, Taiwan. [60]Division of General Surgery, Department of Surgery, School of Medicine, College of Medicine, Taipei Medical University, Taipei, Taiwan. [61]Division of Gastroenterology, Department of Internal Medicine, Kaohsiung Medical University Hospital, Kaohsiung Medical University, Kaohsiung, Taiwan. [62]Department of Medicine, Faculty of Medicine, College of Medicine, Kaohsiung Medical University, Kaohsiung, Taiwan. [63]Department of Internal Medicine, Kaohsiung Medical University Gangshan Hospital, Kaohsiung, Taiwan. [64]Division of Breast Oncology and Surgery, Department of Surgery, Kaohsiung Medical University Chung-Ho Memorial Hospital, Kaohsiung Medical University, Kaohsiung, Taiwan. [65]Department of Post Baccalaureate Medicine, College of Medicine, Kaohsiung Medical University, Kaohsiung, Taiwan. [66]Department of Surgery, Faculty of Medicine, College of Medicine, Kaohsiung Medical University, Kaohsiung, Taiwan. [67]Sleep Medicine Center, Department of Neurology, Kaohsiung Medical University Hospital, Kaohsiung Medical University, Kaohsiung, Taiwan. [68]Department of Neurology, College of Medicine, Kaohsiung Medical University, Kaohsiung, Taiwan. [69]Division of Nephrology, Department of Internal Medicine, Kaohsiung Medical University Hospital, Kaohsiung Medical University, Kaohsiung, Taiwan. [70]Department of Internal Medicine, Kaohsiung Municipal Siaogang Hospital, Kaohsiung Medical University, Kaohsiung, Taiwan. [71]Faculty of Medicine, College of Medicine, Kaohsiung Medical University, Kaohsiung, Taiwan. [72]Department of Urology, Kaohsiung Medical University Hospital, Kaohsiung Medical University, Kaohsiung, Taiwan. [73]Department of Urology, Kaohsiung Medical University Gangshan Hospital, Kaohsiung, Taiwan. [74]Hepatobiliary Division, Department of Internal Medicine, Kaohsiung Medical University Hospital, Kaohsiung Medical University, Kaohsiung, Taiwan. [75]Faculty of Internal Medicine and Hepatitis Research Center, School of Medicine, College of Medicine, Kaohsiung Medical University, Kaohsiung, Taiwan. [76]Division of Pulmonary and Critical Care Medicine, Department of Internal Medicine, Kaohsiung Medical University Hospital, Kaohsiung Medical University, Kaohsiung, Taiwan. [77]Department of Internal Medicine, School of Medicine, College of Medicine, Kaohsiung Medical University, Kaohsiung, Taiwan. [78]Department of Internal Medicine, Kaohsiung Municipal CiJin Hospital, Kaohsiung, Taiwan. [79]Orthopaedic Research Center and Department of Orthopedics, College of Medicine, Kaohsiung Medical University, Kaohsiung, Taiwan. [80]Regenerative Medicine and Cell Therapy Research Center, Kaohsiung Medical University, Kaohsiung, Taiwan. [81]Department of Orthopedics, Kaohsiung Municipal Ta-Tung Hospital and Kaohsiung Medical University Hospital, Kaohsiung, Taiwan. [82]Department of Neurology, Cognition and Aging Center, Kaohsiung Chang Gung Memorial Hospital, Kaohsiung, Taiwan. [83]Institute for Translational Research in Biomedicine, Kaohsiung Chang Gung Memorial Hospital, Kaohsiung, Taiwan. [84]School of Medicine, College of Medicine, National Sun Yat-sen University, Kaohsiung, Taiwan. [85]Center for Artificial Intelligence in Medicine, Chang Gung Memorial Hospital, Taoyuan, Taiwan. [86]Division of Rheumatology, Allergy and Immunology, Chang Gung Memorial Hospital, Taoyuan, Taiwan. [87]Department of Internal Medicine, College of Medicine, Chang Gung University, Taoyuan, Taiwan. [88]Division of Allergy Immunology and Rheumatology, Chang Gung Memorial Hospital, Chiayi, Taiwan. [89]Pediatric Internal Medicine, Department of Internal Medicine, Kaohsiung Chang Gung Memorial Hospital, Taipei, Taiwan. [90]Chang Gung University College of Medicine, Kaohsiung, Taiwan. [91]Division of General Surgery, Department of Surgery, Linkou Chang Gung Memorial Hospital, Taoyuan, Taiwan. [92]Department of Rheumatology, Allergy and Immunology, Linkou Chang Gung Memorial Hospital, Taoyuan, Taiwan. [93]Department of Gastroenterology and Hepatology, New Taipei Municipal TuCheng Hospital, New Taipei City, Taiwan. [94]Department of Gastroenterology and Hepatology, Linkou Chang Gung Memorial Hospital, Taoyuan, Taiwan. [95]Graduate Institute of Clinical Medicine, College of Medicine, Chang Gung University, Taoyuan, Taiwan. [96]Department of Internal Medicine, National Taiwan University Hospital, Taipei, Taiwan. [97]Department of Internal Medicine, National Taiwan University College of Medicine, Taipei, Taiwan. [98]Department of Internal Medicine, National Taiwan University Hospital, Yunlin Branch, Yunlin, Taiwan. [99]Department of Neurology, National Taiwan University Hospital, Taipei, Taiwan. [100]Department of Neurology, National Taiwan University College of Medicine, Taipei, Taiwan. [101]Department of Internal Medicine, National Taiwan University Cancer Center, Taipei, Taiwan. [102]Graduate Institute of Clinical Medicine, National Taiwan University College of Medicine, Taipei, Taiwan. [103]Institute of Medicine, Chung Shan Medical University, Taichung, Taiwan. [104]Department of Medical Research, Chung Shan Medical University Hospital, Taichung, Taiwan. [105]Division of Hematology and Oncology, Department of Internal Medicine, Chung Shan Medical University Hospital, Taichung, Taiwan. [106]School of Medicine, Chung Shan Medical University, Taichung, Taiwan. [107]Endocrinology and Metabolism, Tri-Service General Hospital, National Defense Medical Center, Taipei, Taiwan. [108]Nephrology, Tri-Service General Hospital, National Defense Medical Center, Taipei, Taiwan. [109]Neurology, Tri-Service General Hospital, National Defense Medical Center, Taipei, Taiwan. [110]Pulmonary Medicine, Tri-Service General Hospital, National Defense Medical Center, Taipei, Taiwan. [111]Pediatrics, Tri-Service General Hospital, National Defense Medical Center, Taipei, Taiwan. [112]Department of Medical Research, Tri-Service General Hospital, National Defense Medical Center, Taipei, Taiwan. [113]Cardiovascular and Mitochondria Related Disease Research Center, Hualien Tzu Chi Hospital, Buddhist Tzu Chi Medical Foundation, Hualien, Taiwan. [114]Center of General Education, Buddhist Tzu Chi Medical Foundation, Tzu Chi University of Science and Technology, Hualien, Taiwan. [115]Bioinnovation Center, Buddhist Tzu Chi Medical Foundation, Hualien, Taiwan. [116]Department of Neurosurgery, Hualien Tzu Chi Hospital, Buddhist Tzu Chi Medical Foundation, Hualien, Taiwan. [117]Department of Laboratory Medicine, Hualien Tzu Chi Hospital, Buddhist Tzu Chi Medical Foundation, Hualien, Taiwan. [118]Department of Laboratory Medicine and Biotechnology, Tzu Chi University, Hualien, Taiwan. [119]Department of Dermatology, Hualien Tzu Chi Hospital, Buddhist Tzu Chi Medical Foundation, Hualien, Taiwan. [120]School of Medicine, Tzu Chi University, Hualien, Taiwan. [121]Department of Obstetrics and Gynecology, Far Eastern Memorial Hospital, New Taipei, Taiwan. [122]Graduate School of Biotechnology and Bioengineering, Yuan Ze University, Taoyuan, Taiwan. [123]Department of Obstetrics and Gynecology, National Taiwan University Hospital, Taipei, Taiwan. [124]Department of Orthopedic Surgery, Far Eastern Memorial Hospital, New Taipei City, Taiwan. [125]Department of Family Medicine, Far Eastern Memorial Hospital, New Taipei City, Taiwan. [126]Department of Genomic Medicine and Center for Medical Genetics, Changhua Christian Hospital, Changhua, Taiwan. [127]Department of Cardiology, Fu Jen Catholic University Hospital, Fu Jen Catholic University, New Taipei City, Taiwan. [128]School of Medicine, College of Medicine, Fu Jen Catholic University, New Taipei City, Taiwan. [129]Division of Nephrology, Department of Internal Medicine, Fu Jen Catholic University Hospital, Fu Jen Catholic University, New Taipei City, Taiwan. [130]Koo Foundation Sun Yat-Sen Cancer Center, Taipei, Taiwan. [131]Division of Urology, Taipei City Hospital Ren Ai Branch, Taipei, Taiwan. [132]University of Taipei, General Education Center, Taipei, Taiwan. [133]Department of Urology, College of Medicine and Shu-Tien Urological Research Center, National Yang Ming Chiao Tung University, Taipei, Taiwan. [134]Division of Gastroenterology and Hepatology, Department of Internal Medicine, Ditmanson Medical Foundation Chia-Yi Christian Hospital, Chiayi City, Taiwan. [135]Clinical Trial Center, Department of Medical Research, Ditmanson Medical Foundation Chia-Yi Christian Hospital, Chiayi City, Taiwan. [136]School of Medicine, Fu-Jen Catholic University, New Taipei City, Taiwan. [137]Department of Clinical Pathology, Cathay General Hospital, Taipei, Taiwan. [138]Department of Internal Medicine, Cathay General Hospital, Taipei, Taiwan. [139]Department of Anesthesiology, Cathay General Hospital, Taipei, Taiwan. [140]Department of Urology, Fu Jen Catholic University Hospital, Fu Jen Catholic University, New Taipei City, Taiwan. [141]Department of Medical Research, Far Eastern Memorial Hospital, New Taipei City, Taiwan. [142]Graduate Institute of Medicine and Graduate Program in Biomedical Informatics, Yuan Ze University, Taoyuan, Taiwan. [143]Division of Rheumatology, Immunology and Allergy, Tri-Service General Hospital, Taipei, Taiwan. [144]Cardiovascular Research Institute, Institute for Human Genetics, and Department of Dermatology, University of California, San Francisco, CA, USA. [145]These authors contributed equally:

Chun-Yu Wei, Ming-Shien Wen, Chih-Kuang Cheng, Yi-Jing Sheen, Tsung-Chieh Yao, Sing-Lian Lee. [146]These authors jointly supervised this work: Chien-Hsiun Chen, Chih-Cheng Chien, Han-Sun Chiang, Yen-Ling Chiu, Hsiang-Cheng Chen, and Pui-Yan Kwok. ✉e-mail: nancy.wei324@gmail.com; chchen@ibms.sinica.edu.tw; chiencmail@gmail.com; a00001@mail.fjuh.fju.edu.tw; yenling.chiu@gmail.com; alex0624x@yahoo.com.tw; Pui.Kwok@ucsf.edu

