## [Peer Review file · Nature Communications]

Clinical Impact of Pharmacogenetic Risk Variants in a Large Chinese Cohort

Corresponding Author: Professor Pui-Yan Kwok

Version 0:

Reviewer comments:

Reviewer #1

(Remarks to the Author)

This very relevant population pharmacogenetics study comprises more than 450000 individuals, 19 pharmacogenes, and around 3000 PGx markers in a non-European cohort. Moreover, four gene-drug pairs were more extensively analyzed in this Han Chinese cohort. It is well-written, and the analyses are very interesting and new.

Abstract:

There is a statement, "The largest Han Chinese cohort," but the number of individuals is not specified. Please insert the number of studied individuals in the abstract, which is one of the strengths of the study. I suggest specifying which 4 gene-drug pairs were studied in the abstract.

Introduction:

Please be more specific about the four gene-drug pairs analyzed (the genes are not cited) and justify why you have chosen only these four pairs if the whole genome sequence is available.

Results:

Table S3: It is unclear why the results are presented only for 249796 individuals if the total N is almost 500000. I suggest inserting the N in line 34 to clarify this information.

Throughout the whole text, the gene names are not in italics. Please revise carefully.

Table 1: Instead of alphabetic order, I suggest classifying the drugs in decreasing values from the "Total" column. As it is, it is difficult to identify the patterns described in the text.

Line 175: Table s5 should be cited in this section. It is cited in the next section. Please correct.

Line 242: The citation should be Table s6 instead of Table s5

Line 286: 7 individuals treated with six types of statins is curious information. The cited table needs to bring this information. I suggest detailing better these different treatments because they can influence results.

Lines 286 and 290: The only supplementary table comprising statins is Table s7. Please revise these citations.

Line 297: Table s7 does not present the cited information.

Line 303: Table s9 does not present the cited information.

Lines 333 and 335: These tables do not exist. Table s9 is the only with NSAID information.

Discussion:

I missed more comparisons with other PGx studies comprising other Asian populations. Please insert.

Methods:

Line 430: It is not clear how many SNPs are included in the original TPMI array. Please insert.

Line 431: What are TPM (I think it should be TPM1) and TPM2 arrays? Even though the number of SNPs is presented in the results, it should appear in the methods section.

Line 488: The variants of CYP2B6 gene are not specified. Please specify.

Lines 493-494: "We only kept the individual with..." - it is unclear if you are referring to the people or the drugs. Please rewrite this sentence to clarify.

Lines 539-549: Please specify the software used for the statistical analyses.

Reviewer #2

(Remarks to the Author)

Thank you for providing this important work

Some things you may want to consider

- 1) Line 42 - should outcome be outcomes?
- 2) Throughout the manuscript there is inconsistency in terminology used - consider standardizing
 - a. adverse reactions vs side effects vs adverse effects
 - b. medications vs drugs
 - c. prescription guidelines vs clinical practice recommendations
 - d. ADRs vs ADEs
 - e. US FDA vs FDA
- 3) You should spell out the abbreviation then add the abbreviation in () see lines 75 and 76
- 4) once you have added an abbreviation be sure to use for remainder of manuscript - see line 102 where pharmacogenetics is spelled out
- 5) consider using "clinicians" instead of "providers" to be inclusive
- 6) It is best to not start a sentence with a number unless it is spelled out - see lines 134 and 136
- 7) line 50 - sentence beginning with "Here we show..." - I think there may be a missing "and" between commas
- 8) line 62 - I would change to say effectiveness and/or safety
- 9) line 78 - I am not certain we can state that the FDA has advocated for incorporating preemptive PGx testing into routine care - needs a reference if you feel this is an accurate statement
- 10) line 82 - should this be reference # 7 or 8?
- 11) line 83 uses PG vs PGx
- 12) there is reference to drug prescriptions - some of the meds are available as over the counter medications so maybe refer to medication use history vs prescriptions
- 13) standardize to have gene names in italics
- 14) you mention top 20 drugs and high risk drugs - please define somewhere
- 15) Table 1 - caution is spelled incorrectly; consider delineating the top 20 and high risk meds in the table somehow
- 16) sentence that starts on line 188 and ends on 191 - consider making 2 sentences
- 17) in figures where you use red for significant associations - can you make the lines bolder to be able to see better?
- 16) line 239 - I don't think "most of" makes sense here
- 17) line 240 - AZA-induces adverse reactions (ADRS) but you are missing the word "drug"
- 18) standard language for extensive metabolizers is now "normal metabolizers" - consider updating
- 19) check tense throughout - this was retrospective so should be past
- 20) line 282 and 286 - reference to "types of statins" - suggest change to "statin agent" or "statin prescription"
- 21) does severe form of statin-associated muscle events need to be defined in the Figure 5 as Rhabdo?
- 22) line 322-326 should have a reference or should be removed
- 23) line 333-335 should have a reference or should be removed
- 24) sentence beginning in 347 ending in 349 does not seem to make sense - not sure what you are trying to state
- 25) sentence beginning in line 354 ending in 356 - there would be no recommendations to change a dose of 10-20 mg so this sentence should simply state The post prescribed and effective dose of simvastatin in Taiwan ranges from 10-20 mg.
- 26) I don't think Firstly and thirdly should be used
- 27) Lines 365-370 seems pretty controversial - PGx is to be used as an additional piece of information and not an end all be all - I think it would be good to highlight the utility of PGx information is in addition to other clinical factors
- 28) Consider providing examples from your patients of the reversible, non-life threatening events (line 367) and times when there is no equally effective drug in many cases (368) and add references to these statements
- 29) lines 372-380 - please note the CPIC and FDA PGx guidance often does evaluate and provide risk mitigation management strategies
- 30) did you do NGS or WGS - refers to both in the manuscript
- 31) did you account for phenocoverion?
- 32) line 436 -should "image" be "imaging"?
- 33) consider adding the weblinks tot he references vs in the text of the manuscript
- 34) add the date accessed to the web links
- 35) add references to the CPIC and FDA guidance specific to the drug-gene combinations
- 36) line 500 - how did you measure compliance?
- 37) line 523 - celecoxib does not need to be capitalized
- 38) line 525-531 - consider adding more explanation of what the numbers/letters are in the () CPT codes or ?
- 39) line 501 - how did you define several allergic reactions?

Reviewer #3

(Remarks to the Author)

In this manuscript Wei et al. leverage data from the large Taiwan Precision Medicine Initiative and report on the overall prevalence of genetic variants known to modulate pharmacogenetic outcomes.

The results are interesting for they provide a window into a still under-represented population. They report on the presence of actionable variants in 19 genes that can affect the response of 58 commonly prescribed medications. In addition,

derivation of empirical prescription events and drug dosages as exposed in Table 1 can be a valuable resource for future studies on the cost-effectiveness of the implementation of PGx programs within the Taiwanese population.

Unfortunately, the reach of their analysis is limited. They were only able to access the association to 4 PGx gene-drug pairs and, despite the very large sample size, by using a retrospective design with data captured through EMR could not derive reliable estimates of the occurrence of side effects or adverse reactions. In addition, as would be expected with such a study design, they were not able to access the potential for improved outcomes based on PGx information. In result, I believe the results are noteworthy for a specific audience, but are not necessarily original, for the authors reproduce results already well-known in the literature.

I also have some technical concerns.

Probably the most unclear section of the manuscript is the extraction and validation of data for Clopidogrel, AZA, statins and NSAIDs (the 4 PGx gene-drug pairs). Each of these cohorts/studies should be detailed and the data extraction process and validation should be thoroughly described. Because the data is retrospective, it is unclear how the authors are dealing with temporality issues, drug dosages, missing or incomplete information from the EMR, multi-drug prescription, and adherence. It is also unclear what is the sensitivity and specificity of the codes used for extraction within the different medical systems that participated in the study. The same issues also apply to the extraction of AZA adverse reactions from the EMR. How was this data extracted and what are the positive and negative predictive values for these features? Inconsistencies and measurement error may account for most of the difference between the observed results and results from other studies. In other words, these very same problems bring into question the accuracy of the most important conclusion of the manuscript on this section, which is that the clinical impact of the PGx information is limited.

One important point that was not clear from the manuscript is whether the PGx variants analyzed were genotyped or sequenced. From the Methods section is not clear. In the Data Source section it is stated that samples were genotyped and later imputed. However, in the Pharmacogenomic variants analysis section it is stated that "the SNP genotype and imputation data of passed variants were validated by NGS, Sanger sequencing, and Sequenom MassARRAY platform". Did the authors really confirmed all actionable alleles by 4 different technologies for 500,000 individuals?? Or this was just a validation of the accuracy of the genotyped and imputed variants within a limited number of samples? Please clarify.

Another minor point is that authors should also consider deriving estimates for models that assume a recessive model for some of the PGx variants studied. It is well described that for some of the variants the effect estimates among different genotypes are preponderant at the homozygous state.

Version 1:

Reviewer comments:

Reviewer #1

(Remarks to the Author)

This is a very relevant population pharmacogenetics study, which comprises more than 450000 individuals, 19 pharmacogenes, and around 3000 PGx markers in a non-European population. Moreover, four gene-drug pairs were more extensively analyzed in this Han Chinese cohort. It is well-written, and the analyses are very interesting and new. The methods support the conclusions, and the data are robust enough to meet the expected standards and to be reproduced in other cohorts. The answers to the review concerns and the corrections in the manuscript are fine and have improved substantially the document.

Reviewer #2

(Remarks to the Author)

Thank you for revising and resubmitting

Abstract:

states A key advantage of PGx-guided therapy is the avoidance of adverse events
"avoidance" is a strong word. I think it is more appropriate to state One of the advantages of PGx-guided therapy is to decrease the likelihood of adverse events.

I am not sure what "repositioning" means here

PGx also gains increasing attention in pharmaceutical industry for its potential in drug development and drug repositioning4 .

Intro:

~10 - remove symbol and add text

These sentences seem out of order

Due to limitations in CYP2D6 genotyping, as discussed below, several important gene-drug pairs could not be analyzed. By focusing on these pairs, we ensured both clinical relevance and robust methodological design

Results:

Figure 1a does not have X and y titles but 1b does - make consistent

Define what the actual numbers/% are here vs "most"

However, based on the available medical data, most patients did not suffer from any of the predicted adverse events

Consider changing non-LoF alleles to no LoF alleles

Patients taking clopidogrel with either one or two CYP2C19 LoF alleles were significantly associated with MACE compared to those with non-LoF alleles ($P = 2.97 \times 10^{-27}$, OR = 1.53, 95% CI = 1.42–1

EM vs NM - there are still some references to EM

NUDT15 EM and 17.8% of TPMT EM suffered from AZA-induced ADEs

Methods

should this be TPM1 for the array? Seems to be referred to differently throughout TPM and TPM2 array, genome501 wide imputation data, and HLA imputation data. TPM (686,463 SNPs) and TPM2 (743,227 SNPs) arrays are designed by TPM1 and Thermo Fisher Scientific for the TPM1 project, specifically for the Han Chinese population with superior coverage for GWAS-grid and previously published health related variants.

Reviewer #3

(Remarks to the Author)

Wei et al. carefully provide answers and more detail into their work. I appreciate the comprehensive answers and the work put into the revision of the manuscript. Their responses are clear and acknowledge all the limitations raised in the first revision. As a result, their answers highlight the problems involved with such large-scale analysis and the difficult mission to quantify the overall and specific effects of pharmacogenetic markers. The team should be commended on such enormous amount of work. However, the same facts also highlight the limitation in deriving the effect estimates that are provided. In particular, my main reservation is not regarding their conclusion, but with the novelty of their findings. The question of whether "if one uses the PGx guidelines based on widely accepted genetic risk variant alone, what is the clinical impact?" has been approached by others previously, with similar results overall. I should point, however, the interpretation of the observed results still presents itself with great heterogeneity within the community (while some authors concentrate on the individual-level predictive effect, others are more preoccupied with the sample or population cost-effectiveness of the approach).

Of course, this does not imply on the correctness of the used approach (which I agree was as good as possible considering the scale, scope, and resources available).

REVIEWER COMMENTS

Reviewer #1 (Remarks to the Author):

This very relevant population pharmacogenetics study comprises more than 450000 individuals, 19 pharmacogenes, and around 3000 PGx markers in a non-European cohort. Moreover, four gene-drug pairs were more extensively analyzed in this Han Chinese cohort. It is well-written, and the analyses are very interesting and new.

Response: We thank the reviewer's recognition of the relevance and the novel analyses of our study. We appreciate the good suggestions provided, and as detailed below, we have taken the reviewer's comments and suggestions to heart and revised the manuscript accordingly.

Abstract:

There is a statement, "The largest Han Chinese cohort," but the number of individuals is not specified. Please insert the number of studied individuals in the abstract, which is one of the strengths of the study.

Response: We have revised the Abstract to include the sample size (486,956 individuals), highlighting this as a strength of our study.

Here is the revised sentence (lines 44-47):

"To evaluate the clinical impact of PGx risk variants, we performed a retrospective study using genetic and clinical data from the largest Han Chinese cohort, **comprising 486,956 individuals**, assembled by the Taiwan Precision Medicine Initiative."

I suggest specifying which 4 gene-drug pairs were studied in the abstract.

Response: We have revised the Abstract to specify the four gene-drug pairs studied.

Here is the revised sentence (lines 49-52):

"Here we show the detailed analyses of four gene-drug pairs, **azathioprine (NUDT15/TPMT), clopidogrel (CYP2C19), statins (ABCG2/CYP2C9/SLCO1B1), and NSAIDs (CYP2C9)**, for which sufficient data exists for statistical power."

Introduction:

Please be more specific about the four gene-drug pairs analyzed (the genes are not cited) and justify why you have chosen only these four pairs if the whole genome sequence is available.

Response: The four drug-gene-outcome pairs analyzed in this study were *NUDT15/TPMT* and azathioprine-induced myelosuppression, *CYP2C19* and clopidogrel-related major cardiovascular events (MACEs), *ABCG2/CYP2C9/SLCO1B1* and statin-associated myopathy (SAMs), and *CYP2C9* and NSAID-linked gastrointestinal (GI) and renal toxicity.

These pairs were selected based on several considerations. First, they were the ones with sufficient sample size for statistical power, where large numbers of PGx variant carriers and corresponding clinical outcomes were found in the TPMI cohort. This allowed us to perform meaningful and statistically powerful analyses, underscoring the TPMI cohort's capability to study clinically impactful PGx associations. Second, relevant clinical data were available for these gene-drug pairs. Other gene-drug pairs could not be evaluated due to incomplete clinical records. For example, assessing *CYP2C19* and proton pump inhibitors (PPIs) requires additional information, such as pre- and post-treatment upper gastrointestinal endoscopy results, imaging reports, or *Helicobacter pylori* testing that are not available in the TPMI database. As another example, the complex genetic structure of *CYP2D6* precludes accurate determination of diplotypes and phenotypes using the current TPMI genotype dataset based on genotyping data. As a result, analyses involving drugs significantly influenced by *CYP2D6* cannot be performed and are excluded. Efforts are actively underway to address this limitation by developing a dedicated *CYP2D6* genotyping pipeline, which will enable future studies to include this critical gene.

Regarding whole-genome sequencing (WGS), it was performed for only a small subset of participants (1,498 individuals) for validation of the SNP array genotyping pipeline and building an imputation dataset. The number is too small for meaningful analysis. For this study, we relied on SNP array data (TPM and TPM2 arrays) for all participants. This approach allows us to maximize cohort size with high-quality genotyping that has been validated through WGS.

By focusing on these four carefully selected gene-drug pairs, we demonstrated the TPMI cohort's ability to perform clinically meaningful and methodologically rigorous PGx analyses. Importantly, these findings showcase the potential of TPMI to contribute to actionable insights in personalized medicine. As the TPMI database continues to integrate more comprehensive clinical and genetic data, including enhanced *CYP2D6* genotyping and expanded clinical records, we anticipate conducting broader analyses that will further extend the impact and scope of this work.

We have revised the manuscript to clarify these points and include the specific gene-drug pairs analyzed.

Here are the revised sentences (lines 101-112):

“Specifically, we evaluated the association between genetic variants and risk for adverse events, including *NUDT15/TPMT* and azathioprine (AZA)-induced myelosuppression, *CYP2C19* and clopidogrel-related major adverse cardiovascular events (MACEs), *ABCG2/CYP2C9/SLCO1B1* and statin-associated myopathy (SAMs), and *CYP2C9* and non-steroidal anti-inflammatory drugs (NSAID)-linked gastrointestinal (GI) and renal toxicity. **These pairs were selected based on sufficient sample size for statistical power, and the availability of relevant clinical data. Due to limitations in *CYP2D6* genotyping, as discussed below, several important gene-drug pairs could not be analyzed. By focusing on these pairs, we ensured both clinical relevance and robust methodological design.**”

Results:

Table S3: It is unclear why the results are presented only for 249796 individuals if the total N is almost 500000. I suggest inserting the N in line 34 to clarify this information.

Response: The number 249,796 represents participants who were not prescribed any high-risk PGx drugs, as indicated in the first row of the table. The total number of participants in the TPMI cohort is 486,956, which includes both those prescribed high-risk PGx drugs and those unexposed. To address your concern, we have updated Supplementary Table 3 to label the first row as “0 (unexposed)” and added a “SUM = 486,956” row at the bottom to clearly indicate the total cohort size. These changes improve the clarity of the table

and align with your suggestion.

Here is the revised Supplementary Table 3:

Number of drug prescribed per person	Case number	Accumulated count	Accumulated %
0 (unexposed)	249,796		
1	98,724	237,160	48.7%
2	60,111	138,436	28.4%
3	33,926	78,325	16.1%
4	19,251	44,399	9.1%
5	10,596	25,148	5.2%
6	6,114	14,552	3.0%
7	3,550	8,438	1.7%
8	2,124	4,888	1.0%
9	1,133	2,764	0.6%
10	673	1,631	0.3%
11	440	958	0.2%
12	242	518	0.1%
13	132	276	0.1%
14	78	144	0.0%
15	42	66	0.0%
16	14	24	0.0%
17	9	10	0.0%
18	1	1	0.0%
SUM	486,956		

Throughout the whole text, the gene names are not in italics. Please revise carefully.

Response: We have carefully reviewed the manuscript and revised all gene names to be italicized throughout the text, in accordance with standard conventions.

Table 1: Instead of alphabetic order, I suggest classifying the drugs in decreasing values from the “Total” column. As it is, it is difficult to identify the patterns described in the text.

Response: We agree that for Table 1, arranging the drugs in decreasing order based on the “Total” column improves clarity and facilitates the identification of the described patterns. We have made the modification as requested and updated the table accordingly.

Here is the revised Table 1:

High-risk drugs	Total	Standard treatment	Increased dose	Decreased dose	Alternative	Enzyme activity test	Used in caution	Indetermined
Atorvastatin	70,273	55,516		14,752				5
Celecoxib	50,200	46,570		3,608				22
Rosuvastatin	46,617	17,019		5,263			24,320	15
Lansoprazole	44,378	17,063	284	26,955				76
Clopidogrel	33,664	13,101		20,517				46
Omeprazole	33,407	12,807	210	20,344				46
Pitavastatin	30,876	24,284		6,591				1
Flurbiprofen	28,989	26,807		2,159				23
Pantoprazole	28,138	10,889	185	17,024				40
Gentamicin	23,953	23,816			109			28
Dexlansoprazole	19,147	7,387		11,604	122			34
Piroxicam	13,933	12,841		1,061	17			14
Simvastatin	10,230	7,973		2,256				1
Meloxicam	10,174	9,447		715	7			5
Sulfasalazine	9,777	7,679					2,082	16
Imipramine	9,636	8,195			1,420			21
Azathioprine	8,486	6,602		1,774	72			38
Sulfamethoxazole and Trimethoprim	7,699	6,016					1,666	17
Pravastatin	6,955	5,510		1,445				-
Citalopram	6,942	2,714		4,160	51			17
Escitalopram	6,845	2,677		4,100	51			17
Doxepin	6,773	5,745			1,018			10
Ibuprofen	6,681	6,135		541				5
Warfarin	6,111	1,203		4,905				3
Allopurinol	5,251	4,373			878			
Sertraline	4,786	3,895		868	12			11
Tobramycin	4,190	4,168			16			6
Lovastatin	3,563	2,784		779				-
Fluvastatin	3,526	2,548		968	5			5
Tacrolimus	3,299	1,668	1,630					1
Isoniazid	2,918	2,269					646	3
Oxcarbazepine	2,229	2,056			173			
Ribavirin	1,873	1,591					282	
Carbamazepine	1,319	1,210			109			
Clobazam	1,283	484		799				-
Abacavir	692	691			1			
Phenytoin	670	574		42	54			-
Amikacin	635	631			3			1
Irinotecan	596	324		264				8
Peginterferon alfa-2a	576	504					72	
Voriconazole	343	292			51			
Efavirenz	258	179		79				-
Nateglinide	227	227		-				-
Peginterferon alfa-2b	218	181					37	
Pazopanib	120	120						
Dapsone	108	96			2	5		5
Nilotinib	73	66					6	1
Streptomycin	59	59			-			-
Atazanavir	56	50			6			-
Mercaptopurine	37	30		6	-			1
Rasburicase	31	29			1	-		1
Brivaracetam	28	25		3				-
Lapatinib	27	27						
Tenoxicam	23	23		-	-			-
Paromomycin	15	15			-			-
Abrocitinib	11	9		2				-
Nitrofurantoin	5	4			-	1		-
Kanamycin	5	5			-			-

Line 175: Table s5 should be cited in this section. It is cited in the next section. Please correct.

Response: We have revised the manuscript to include the citation for Supplementary Table 5 in the appropriate section, as requested, which ensures the accuracy and clarity of the manuscript.

Here are the revised sentences (lines 201-213):

" Patients taking clopidogrel with either one or two *CYP2C19* LoF alleles were significantly associated with MACE compared to those with non-LoF alleles ($P = 2.97 \times 10^{-27}$, OR = 1.53, 95% CI = 1.42–1.65), including MI ($P = 2.09 \times 10^{-31}$, OR = 3.98, 95% CI = 3.15–5.02), UA ($P = 7.24 \times 10^{-13}$, OR = 1.88, 95% CI = 1.58–2.23), HF ($P = 1.77 \times 10^{-4}$, OR = 1.37, 95% CI = 1.22–1.55), TVR ($P = 1.91 \times 10^{-4}$, OR = 1.29, 95% CI = 1.13–1.47), and stroke ($P = 6.31 \times 10^{-5}$, OR = 1.41, 95% CI = 1.19–1.67) under multivariate analysis (Fig. 3 and **Supplementary Table 5**). Both *CYP2C19* intermediate metabolizers (IM) and poor metabolizers (PM) had higher MACE incidence under clopidogrel treatment; however, 85.9% of people with *CYP2C19* LOF alleles tolerated clopidogrel treatment well. There was no significant difference in risk of CV death under clopidogrel therapy for those with or without *CYP2C19* LoF alleles (Fig. 3 and **Supplementary Table 5**)."

Line 242: The citation should be Table s6 instead of Table s5

Response: We have corrected the citation to refer to Supplementary Table 6 instead of Supplementary Table 5 in the revised manuscript.

Here is the revised sentence (line 242-246):

"The distribution of sex, comorbidity, and usage of concurrent drugs (aspirin, allopurinol, corticosteroids, and methotrexate) were not significantly different in the patients with AZA-induced adverse drug events (ADEs) compared with AZA tolerant controls, but concomitant use of allopurinol and AZA increased risk for adverse events development (Supplementary Table 6)."

Line 286: 7 individuals treated with six types of statins is curious information. The cited table needs to bring this information. I suggest detailing better these different treatments because they can influence results.

Response: As suggested, we have added a new supplementary table (Supplementary Table 8: Statin users with muscle complaints toward different types of statins) to provide detailed information on individuals treated with multiple statins. To address your concern, we have also revised the description to clarify the reasons for drug switching, particularly among the seven patients treated with six types of statins

Here are the revised sentences (lines 290-301):

“Of these participants, 34,411 participants took more than one statin agent, and 7 were treated with 6 different statin agents (Supplementary Table 8). The switching between statins in these rare cases was attributed to reasons such as intolerance and inefficacy. Specifically, these seven patients experienced adverse events with certain statins, such as muscle toxicity, prompting switches to alternative statins. In some cases, the next statin was well-tolerated but failed to adequately control LDL levels, necessitating further changes. Ultimately, all seven patients were transitioned to non-statin therapies, including restricted diet control. Although statins shared similar structures, many participants who suffered from muscle toxicity caused by one statin could take another statin without any adverse events (Supplementary Table 8).”

Here is the added Supplementary Table 8:

Supplementary Table 8: Statin users with muscle complaints toward different types of statin							
The type of statins prescribed	SAM with how many statins						
	0	1	2	3	4	5	6
1	92064	722	0	0	0	0	0
2	24925	856	49	0	0	0	0
3	6388	428	107	24	0	0	0
4	1222	133	50	17	5	0	0
5	141	29	15	6	9	0	0
6	2	1	1	2	0	1	0

Lines 286 and 290: The only supplementary table comprising statins is Table s7. Please revise these citations.

Response: We have updated the citations accordingly. In this revised version of the manuscript, due to the addition of a new supplementary table (Supplementary Table 8: Statin users with muscle complaints toward different types of statin), the table comprising statins is now Supplementary Table 9.

Here are the revised sentences (lines 298-303):

“Although statins shared similar structures, many participants who suffered from muscle toxicity caused by one statin could take another statin without any adverse events (Supplementary Table 8). Therefore, genetic associations with myopathy induced by different statins were analyzed separately (Fig. 5 and Supplementary Table 9).”

Line 297: Table s7 does not present the cited information.

Response: In the original manuscript, the table comprising statins was incorrectly cited as Supplementary Table 7, but it should have been Supplementary Table 8. In this revised manuscript, due to the addition of a new supplementary table, the table comprising statins is now correctly referenced as Supplementary Table 9.

Here is the revised sentence (lines 305-310):

“Our study showed that people with poor function of *ABCG2* had higher risk of myositis when taking atorvastatin ($P = 3.58 \times 10^{-3}$, OR = 2.27, 95% CI = 1.31–3.94) or simvastatin ($P = 1.26 \times 10^{-2}$, OR = 3.43, 95% CI = 1.3–9.05); whereas people with *ABCG2* decreased function phenotype had higher risk of myalgia while taking fluvastatin ($P = 5.39 \times 10^{-3}$, OR = 1.97, 95% CI = 1.22–3.18) (Fig. 5a-c and Supplementary Table 9).”

Line 303: Table s9 does not present the cited information.

Response: In the original manuscript, the table summarizing statin-associated risks was incorrectly cited as Supplementary Table 9 when it should have been Supplementary Table 8. In the revised manuscript, due to the addition of a new supplementary table, this table is now correctly referenced as Supplementary Table 9. We have updated all relevant citations accordingly in the revised manuscript (lines 314-319).

“Furthermore, although the pharmacokinetic of fluvastatin was known to be affected by *CYP2C9* phenotypes, the frequency of *CYP2C9* LOF was not significantly higher in patients who experienced SAM after receiving fluvastatin in our cohort ($P = 8.63 \times 10^{-1}$, OR = 0.93, 95% CI = 1.22–3.18) (Fig. 5f and Supplementary Table 9).”

Lines 333 and 335: These tables do not exist. Table s9 is the only with NSAID information.

Response: In the revised manuscript, due to the addition of a new supplementary table (Supplementary Table 8), the numbering of the tables has changed. The table comprising information about NSAID-induced side effects and their associations with comorbidities and concurrent medications is now correctly cited as Supplementary Table 10. Additionally, a new supplementary table (Supplementary Table 11) and a new figure (Fig. 6) have been added to present the influence of CYP2C9 phenotypes on NSAID-associated adverse events, including ADRs, GI events, and kidney-related outcomes.

Here is the revised sentence (lines 345-347):

“We found that those with the *CYP2C9* PM phenotype were predisposed to NSAID-induced upper GI bleeding, although the sample size is relatively small (Fig. 6 and Supplementary Table 11).”

Here is the added Fig. 6 and figure legend:

a. CYP2C9 LOF

b. CYP2C9 IM

c. CYP2C9 PM

Fig. 6. Influence of CYP2C9 in NSAID-associated adverse events

Forest plot of the ADE risk in statin users with different *CYP2C9* phenotypes. The significant associations were in red. GI = GI discomfort; IM = intermediate metabolizer; LoF = loss-of-function; NM = normal metabolizer; PM = poor metabolizer.

Here is the added Supplementary Table 11:

Supplementary Table 11: Influence of ABCG2, CYP2C9 and SLCO1B1 for statin-associated myopathy									
NSAID users with decreased or poor function CYP2C9 vs with normal function CYP2C9									
Univariate analysis					Multivariate analysis				
ADE	OR	Lower95%CI	Upper95%CI	Pvalue	OR	Lower95%CI	Upper95%CI	Pvalue	
ADE	0.75	0.51	1.09	0.13	0.75	0.51	1.1	0.14	
GI	0.8	0.48	1.34	0.41	0.8	0.48	1.34	0.40	
kidney	0.72	0.49	1.06	0.09	0.72	0.49	1.07	0.10	
NSAID users with decreased function CYP2C9 vs with normal function CYP2C9									
Univariate analysis					Multivariate analysis				
ADE	OR	Lower95%CI	Upper95%CI	Pvalue	OR	Lower95%CI	Upper95%CI	Pvalue	
ADE	0.71	0.48	1.05	0.08	0.71	0.47	1.05	0.09	
GI	0.72	0.42	1.23	0.23	0.71	0.41	1.23	0.23	
kidney	0.71	0.48	1.05	0.08	0.71	0.47	1.05	0.09	
NSAID users with poor function CYP2C9 vs with normal function CYP2C9									
Univariate analysis					Multivariate analysis				
ADE	OR	Lower95%CI	Upper95%CI	Pvalue	OR	Lower95%CI	Upper95%CI	Pvalue	
ADE	2.92	0.65	13.1	0.16	3.37	0.75	15.14	0.11	
GI	6.03	1.34	27.08	0.02	5.97	1.33	26.87	0.02	
kidney	1.35	0.18	10.34	0.77	1.56	0.2	11.93	0.67	

Discussion:

I missed more comparisons with other PGx studies comprising other Asian populations. Please insert.

Response:

To address this concern, we have revised the discussion to better integrate our findings with prior PGx studies with other Asian populations. Our study aligns with recent reports, which underscore the importance of studying population-specific genetic variations in drug response.

Here are the inserted sentences and reference (lines 379-385):

“Our findings, along with previous studies, confirm the prevalence of clinically actionable PGx variants in Asian populations. For instance, studies on Sri Lankan, Indian, South Korean, Thai, and Chinese populations highlight significant differences in the frequency of PGx variants compared to Europeans. Variants like NUDT15, associated with AZA-induced myelotoxicity, and CYP2C19, influencing clopidogrel metabolism, are notably more prevalent in Asians¹⁻⁵.”

Reference:

1. Piriyaongsa, J. et al. Pharmacogenomic landscape of the Thai population from genome sequencing of 949 individuals. *Sci Rep* 14, 30683 (2024). <https://doi.org/10.1038/s41598-024-79018-6>

2. Youn, M. S., Ahn, S. H. & Kim, J. H. Pharmacogenomic profiling of the South Korean population: Insights and implications for personalized medicine. *Front Pharmacol* 15, 1476765 (2024).
<https://doi.org/10.3389/fphar.2024.1476765>
3. Qi, G. et al. Genetic landscape of 125 pharmacogenes in Chinese from the Chinese Millionome Database. *Sci Rep* 11, 19222 (2021).
<https://doi.org/10.1038/s41598-021-98877-x>
4. Sahana, S. et al. Pharmacogenomic landscape of Indian population using whole genomes. *Clin Transl Sci* 15, 866-877 (2022).
<https://doi.org/10.1111/cts.13153>
5. Ranasinghe, P. et al. Frequency of pharmacogenomic variants affecting safety and efficacy of immunomodulators and biologics in a South Asian population from Sri Lanka. *Hum Genomics* 18, 107 (2024).
<https://doi.org/10.1186/s40246-024-00674-w>

Methods:

Line 430: It is not clear how many SNPs are included in the original TPMI array. Please insert.

Line 431: What are TPM (I think it should be TPM1) and TPM2 arrays? Even though the number of SNPs is presented in the results, it should appear in the methods section.

Response: We have inserted that number SNPs (TPM: 686,463 SNPs and TPM2: 743,227 SNPs) in the Method section and added a description of TPM and TPM2 arrays to clarify this issue.

Here is the updated sentence (lines 501-505):

“TPM (686,463 SNPs) and TPM2 (743,227 SNPs) arrays are designed by TPMI and Thermo Fisher Scientific for the TPMI project, specifically for the Han Chinese population with superior coverage for GWAS-grid and previously published health related variants.”

Line 488: The variants of CYP2B6 gene are not specified. Please specify.

Response: We used rs3745274 as a marker to identify decreased or non-functional CYP2B6 alleles. rs3745274 is shared by multiple decreased or non-functional CYP2B6 alleles, including *6, *7, *9, *13, *19, *20, *26, *34, *36, *37, *38, *39, *40, *41, *42, and *43. Due to the complexity of accurately typing each CYP2B6 alleles, which involves intricate SNP combinations and phasing challenges inherent to SNP array datasets, we opted to use

rs3745274 as a surrogate marker rather than identifying individual CYP2B6 allele types. We have updated Supplementary Table 1 to explicitly state that decreased or non-functional CYP2B6 alleles were detected using rs3745274, which is shared by the CYP2B6*6, *7, *9, *13, *19, *20, *26, *34, *36, *37, *38, *39, *40, *41, *42, and *43 alleles.

Lines 493-494: “We only kept the individual with...” - it is unclear if you are referring to the people or the drugs. Please rewrite this sentence to clarify.

Response: We acknowledge that the original sentence was unclear. What we intended to convey is that, when identifying drug users based on prescription data, individuals with only a single prescription record for each drug were excluded. This approach ensures a more accurate analysis of drug usage patterns and treatment outcomes. We have revised this sentence in the manuscript for clarity.

Here is the revised sentence (lines 575-576):

“Individuals with only a single prescription record for each drug were excluded.”

Lines 539-549: Please specify the software used for the statistical analyses.

Response: We have added the information about the software used for statistical analyses to clarify our methods. Statistical analyses were conducted using Python 3.11, and we have updated the manuscript to reflect this.

Here is the added sentence (lines 662-663):

“Statistical analyses were conducted using Python (version 3.11).”

Reviewer #2:

Thank you for providing this important work

Response: We thank the reviewer for recognizing the importance of this work and providing many helpful suggestions. We have revised the paper as suggested.

Some things you may want to consider

1) Line 42 - should outcome be outcomes?

Response: We have revised the text to replace "outcome" with "outcomes" to ensure grammatical accuracy.

Here is the revised sentence (lines 41-43):

“Incorporating pharmacogenetics into clinical practice promises to improve therapeutic **outcomes** by optimizing drug selection and dosage based on genetic factors affecting drug response.”

2) Throughout the manuscript there is inconsistency in terminology used - consider standardizing

a. adverse reactions vs side effects vs adverse effects

b. medications vs drugs

c. prescription guidelines vs clinical practice recommendations

d. ADRs vs ADEs

e. US FDA vs FDA

Response: As suggested, we have carefully reviewed the manuscript to ensure consistency in terminology and have standardized the following terms:

a. Replaced “side effects”, “adverse reactions”, and “adverse effects” with “adverse events”

b. Used “drugs” consistently instead of “medications.”

c. Standardized to “clinical practice recommendations.”

d. Consistently used “ADEs” throughout.

e. Uniformly used “US FDA.”

3) You should spell out the abbreviation then add the abbreviation in () see lines 75 and 76

Response: As suggested, we have revised the text to ensure all abbreviations are spelled out before their first use as you suggested (lines 74-79).

“To facilitate the clinical implementation of PGx, experts from regulatory agencies and consortia such as the **United States Food and Drug Administration (US FDA)** and **Clinical Pharmacogenetics Implementation Consortium (CPIC)** have published clinical practice recommendations based on level of evidence and advocated for incorporating pharmacogenetic testing when a relevant drug is being prescribed.”

4) once you have added an abbreviation be sure to use for remainder of manuscript - see line 102 where pharmacogenetics is spelled out

Response: We have reviewed the manuscript carefully and revised it accordingly to ensure consistency in the use of abbreviations. Regarding original line 102 (line 101 in the revised manuscript), I confirm that the term "pharmacogenetics" is introduced earlier in the manuscript with its abbreviation provided, and it is consistently abbreviated thereafter.

5) consider using "clinicians" instead of "providers" to be inclusive

Response: As suggested, we have revised our paper to use "clinicians" instead of "providers".

Here is the revised sentence (lines 67-70):

"Incorporating PGx into clinical practice is a promising strategy for healthcare **clinicians** to tailor drug selection and dosing to maximize therapeutic benefits while minimizing the risk of drug-related adverse events."

6) It is best to not start a sentence with a number unless it is spelled out - see lines 134 and 136

Response: We have revised the sentences as suggested and have also reviewed the manuscript thoroughly to make sure that no other sentences begin with a number.

Here are revised sentences:

Line 140: "**Among the TPMI participants**, 48.7% of the TPMI participants have been prescribed at least one of the 58 drugs with clinical practice recommendations based on their genetic status in the 19 pharmacogenes."

Line 142: "**Additionally**, 28.4% took two or more drugs with PGx information (Supplementary Table 3)."

7) line 50 - sentence beginning with "Here we show... - I think there may be a missing "and" between commas

Response: Per the reviewer's comment, we reworded the sentence to clarify the point we tried to make.

Here are revised sentences (lines 49-57)

"Here we show the detailed analyses of four gene-drug pairs, azathioprine (*NUDT15/TPMT*), clopidogrel (*CYP2C19*), statins (*ABCG2/CYP2C9/SLCO1B1*), and NSAIDs (*CYP2C9*), for which sufficient data exists for statistical power. **While the results** validate previous findings

that PGx risk variants are significantly associated with drug-related adverse events or ineffectiveness, the excess risk of adverse events or lack of efficacy is small compared to that found in those without the PGx risk variants, and most patients with PGx variants do not suffer from adverse events.”

8) line 62 - I would change to say effectiveness and/or safety

Response: We agree that the suggested change more accurately reflects the potential overlap between these two aspects of drug variability. The revised sentence now reads (lines 62-64):

" Variability in drug effectiveness **and/or** safety greatly impacts therapeutic outcome, with drug-response rates varying widely from 25% to 80% among the commonly used drugs."

9) line 78 - I am not certain we can state that the FDA has advocated for incorporating preemptive PGx testing into routine care - needs a reference if you feel this is an accurate statement

Response: We appreciate your comment and agree that the FDA does not advocate universal preemptive pharmacogenetic testing. We have revised the sentence accordingly to reflect that pharmacogenetic testing is recommended when a relevant medication is being prescribed, rather than as a preemptive measure.

Here is the revised sentence (lines 74-79):

“To facilitate the clinical implementation of PGx, experts from regulatory agencies and consortia such as the United States Food and Drug Administration (US FDA) and Clinical Pharmacogenetics Implementation Consortium (CPIC) have published clinical practice recommendations based on level of evidence and advocated for incorporating **pharmacogenetic testing when a relevant drug is being prescribed.**”

10) line 82 - should this be reference # 7 or 8?

Response: Thank you for catching the original citation error. We have updated the manuscript to include the correct reference with the following citation:
McInnes, G. et al. Pharmacogenetics at Scale: An Analysis of the UK Biobank. Clin Pharmacol Ther 109, 1528-1537 (2021). <https://doi.org/10.1002/cpt.2122>

11) line 83 uses PG vs PGx

Response: We have changed all occurrences of “PG” to “PGx” for consistency throughout the manuscript.

Here is the revised sentence (lines 82-84):

“In Australia, merely 4% of the study participants lacked actionable PGx variants, and 42% of them had more than 2 actionable PGx variants.”

12) there is reference to drug prescriptions - some of the meds are available as over the counter medications so maybe refer to medication use history vs prescriptions

Response: Our study primarily uses prescription records from the TPMI clinical dataset from partner hospitals. These records were used to define drug users and drug usage periods, allowing us to assess adverse events occurring during active medication use. However, we acknowledge that some medications, such as NSAIDs, may also be obtained as over-the-counter products, which could lead to underestimation of their use in our dataset. To address this, we have added the following point to the discussion section as a limitation of the study (lines 440-445).

" Second, some drugs, such as NSAIDs, may be available over the counter (OTC) in Taiwan. Since the TPMI dataset is based on prescription records obtained from hospitals, OTC drug use is not captured. This could lead to underestimation of the use for drugs that are commonly obtained without a prescription, potentially impacting the assessment of drug-related outcomes. "

13) standardize to have gene names in italics

Response: Per the suggestion, we have carefully reviewed the manuscript and revised all gene names to be italicized throughout the text.

14) you mention top 20 drugs and high risk drugs - please define somewhere

Response: As suggested, we have revised the manuscript to explicitly define these terms and updated the wording. We have replaced "top 20 drugs prescribed" with "top 20 most commonly prescribed drugs" throughout the manuscript to clarify that the ranking is based on prescription frequency. We have also defined "high-risk drugs" as those requiring dose adjustments, alternative therapies, or special attention when prescribed to individuals with actionable PGx variants, following US FDA and CPIC recommendations.

Besides, Table 1 has been reordered by the "Total" column in descending order, and the title has been updated to make this clear to the reader. We hope that these revisions ensure the definitions and terms are easy to understand.

Here are the revised sentences (lines 146-152):

"Among the individuals carrying clinically actionable PGx variants, 17.8% of them have been prescribed the responding high-risk drugs, **defined as those requiring dose adjustments, alternative therapies, or additional monitoring when prescribed to individuals with actionable PGx variants** (Fig. 2a and Supplementary Table 2). The **top 20 most commonly prescribed drugs** included statins, NSAIDs, proton-pump inhibitors (PPI), anti-platelet drugs, and antibiotics (Table 1)."

15) Table 1 - caution is spelled incorrectly; consider delineating the top 20 and high risk meds in the table somehow

Response: This has been corrected. Additionally, we have updated Table 1 to better delineate the "Top 20 most commonly prescribed drugs" and "high-risk drugs" as requested. In response, we have updated the column name from "Drug" to "High-risk drugs" to emphasize the focus on drugs requiring caution or adjustments for individuals with actionable PGx variants. Additionally, we have sorted the table by the total number of users to display the top 20 drugs at the bottom part of Table 1.

The "Top 20 most commonly prescribed drugs" are now visually separated in Table 1, with a clear indication of their ranking based on prescription frequency.

Here is the revised Table 1:

High-risk drugs	Total	Standard treatment	Increased dose	Decreased dose	Alternative	Enzyme activity test	Used in caution	Indetermined
Atorvastatin	70,273	55,516		14,752				5
Celecoxib	50,200	46,570		3,608				22
Rosuvastatin	46,617	17,019		5,263			24,320	15
Lansoprazole	44,378	17,063	284	26,955				76
Clopidogrel	33,664	13,101		20,517				46
Omeprazole	33,407	12,807	210	20,344				46
Pitavastatin	30,876	24,284		6,591				1
Flurbiprofen	28,989	26,807		2,159				23
Pantoprazole	28,138	10,889	185	17,024				40
Gentamicin	23,953	23,816			109			28
Dexlansoprazole	19,147	7,387		11,604	122			34
Piroxicam	13,933	12,841		1,061	17			14
Simvastatin	10,230	7,973		2,256				1
Meloxicam	10,174	9,447		715	7			5
Sulfasalazine	9,777	7,679					2,082	16
Imipramine	9,636	8,195			1,420			21
Azathioprine	8,486	6,602		1,774	72			38
Sulfamethoxazole and Trimethoprim	7,699	6,016					1,666	17
Pravastatin	6,955	5,510		1,445				-
Citalopram	6,942	2,714		4,160	51			17
Escitalopram	6,845	2,677		4,100	51			17
Doxepin	6,773	5,745			1,018			10
Ibuprofen	6,681	6,135		541				5
Warfarin	6,111	1,203		4,905				3
Allopurinol	5,251	4,373			878			
Sertraline	4,786	3,895		868	12			11
Tobramycin	4,190	4,168			16			6
Lovastatin	3,563	2,784		779				-
Fluvastatin	3,526	2,548		968	5			5
Tacrolimus	3,299	1,668	1,630					1
Isoniazid	2,918	2,269					646	3
Oxcarbazepine	2,229	2,056			173			
Ribavirin	1,873	1,591					282	
Carbamazepine	1,319	1,210			109			
Clobazam	1,283	484		799				-
Abacavir	692	691			1			
Phenytoin	670	574		42	54			-
Amikacin	635	631			3			1
Irinotecan	596	324		264				8
Peginterferon alfa-2a	576	504					72	
Voriconazole	343	292			51			
Efavirenz	258	179		79				-
Nateglinide	227	227		-				-
Peginterferon alfa-2b	218	181					37	
Pazopanib	120	120						
Dapsone	108	96			2	5		5
Nilotinib	73	66					6	1
Streptomycin	59	59			-			-
Atazanavir	56	50			6			-
Mercaptopurine	37	30		6	-			1
Rasburicase	31	29			1	-		1
Brivaracetam	28	25		3				-
Lapatinib	27	27						
Tenoxicam	23	23		-	-			-
Paromomycin	15	15			-			-
Abrocitinib	11	9		2				-
Nitrofurantoin	5	4			-	1		-
Kanamycin	5	5			-			-

16) sentence that starts on line 188 and ends on 191 - consider making 2 sentences

Response: Per the suggestion, we have revised the sentence by splitting it into two sentences as follows (lines 192-195).

“The demographics, clinical characteristics, and *CYP2C19* status of the patients were found in Supplementary Table 4. **There was** no significant difference in age, gender, and comorbidities between individuals with or without MACE.”

17) in figures where you use red for significant associations - can you make the lines bolder to be able to see better?

Response: We have revised the figures to make the lines representing significant associations bolder for improved visibility and clarity of the figures.

Here is one of the revised figures (Fig. 3)

a. CYP2C19 LOF

b. CYP2C19 IM

c. CYP2C19 PM

16) line 239 - I don't think "most of" makes sense here

Response: We have revised the sentence to improve clarity.

Here is the revised sentence (lines 242-246):

“The distribution of sex, comorbidity, and usage of concurrent drugs (aspirin, allopurinol, corticosteroids, and methotrexate) were not significantly different in the patients with AZA-induced ADEs compared with AZA tolerant controls, but concomitant use of allopurinol and AZA increased risk for adverse events development (Supplementary Table 6).”

17) line 240 - AZA-induces adverse reactions (ADRS) but you are missing the word "drug"

Response: We have revised the text to "AZA-induced ADEs" to address this issue and ensure clarity.

18) standard language for extensive metabolizers is now "normal metabolizers" - consider updating

Response: We have updated the text and figures to align with the current standard terminology by replacing "extensive metabolizers (EM)" with "normal metabolizers (NM)".

19) check tense throughout - this was retrospective so should be past

Response: We have carefully reviewed the manuscript and modified the text to ensure consistent use of the past tense, reflecting the retrospective nature of the study.

20) line 282 and 286 – reference to “types of statins” – suggest change to “statin agent” or “statin prescription”

Response: We have revised the text to use "statin agent" to more accurately describe the individual drugs analyzed in this study (lines 286-292).

" Data from 127,197 TPMI participants who ever treated with any **statin agent** (atorvastatin, fluvastatin, lovastatin, pitavastatin, pravastatin, rosuvastatin, or simvastatin) were analyzed for this study (with individuals previously suffering from muscular disorders excluded). Of these participants, 34,411 participants took more than one **statin agent**, and 7 were treated with 6 different **statin agents** (Supplementary Table 8). "

21) does severe form of statin- associated muscle events need to be defined in the Figure 5 as Rhabdo?

Response: We have updated the figure legend for Figure 5 to clarify the definitions of SAMs and sSAMs.

“Fig. 5. Influence of *ABCG2/SLCO1B1/CYP2C9* in SAMs

Forest plot of the SAM risk in statin users with different *ABCG2*, *SLCO1B1*, or *CYP2C9* phenotypes. The significant associations were in red. IM = intermediate metabolizer; LoF = loss-of-function; NM = normal metabolizer; PM = poor metabolizer. SAM = statin-associated muscle events, including myalgia, myositis, and rhabdomyolysis; sSAM = severe forms of statin-associated muscle events, specifically myositis and rhabdomyolysis.”

22) line 322-326 should have a reference or should be removed

Response: We have revised the manuscript and included the CPIC guideline as a reference to ensure clarity and support (lines 335-340).

“Although substantial evidence links *CYP2C9* deficient phenotype to altered NSAID plasma concentrations, clinical evidence directly substantiating an increased risk of ADEs in individuals with reduced *CYP2C9* metabolism of NSAIDs (such as celecoxib, flurbiprofen, ibuprofen, lornoxicam, meloxicam, piroxicam and tenoxicam) remains limited²⁶.”

Reference:

1. Theken, K. N. et al. Clinical Pharmacogenetics Implementation Consortium Guideline (CPIC) for *CYP2C9* and Nonsteroidal Anti-Inflammatory Drugs. Clin Pharmacol Ther 108, 191-200 (2020). <https://doi.org/10.1002/cpt.1830>

23) line 333-335 should have a reference or should be removed

Response: As suggested, we have revised the sentence in question to make it clear that the observation is based on the findings from Supplementary Table 10.

Here is the revised sentence (lines 347-351):

“It was important to note that comorbidities (such as hypertension, diabetes, and cardiovascular disease) and concurrent drugs (e.g., diuretics, ACE inhibitors) contribute much more significantly to the adverse events of NSAIDs than *CYP2C9* status, as observed in Supplementary Table 10.”

24) sentence beginning in 347 ending in 349 does not seem to make sense - not sure what you are trying to state

Response: The sentence was to point out that inter-population differences in

genetic and non-genetic factors significantly influence drug response, which in turn leads to distinct dosing recommendations between populations, (e.g., such as those applied to Asians and Europeans). We have revised the sentence to improve clarity and integrate it more effectively into the narrative (lines 369-372).

"Well-recognized differences in drug response across populations, driven by genetic and clinical factors, have resulted in distinct optimal dosing recommendations for Asians and Europeans in current clinical practice."

25) sentence beginning in line 354 ending in 356 - there would be no recommendations to change a dose of 10-20 mg so this sentence should simply state The post prescribed and effective dose of simvastatin in Taiwan ranges from 10-20 mg.

Response: We have revised the text and replaced the sentence beginning in line 354 and ending in line 356 (Line 376-379 in the revised manuscript) with the one recommended. Additionally, we have added the sentence, "These observations emphasize the need for expanding PGx studies in underrepresented populations to refine guidelines and improve precision medicine globally," to strengthen the paragraph and highlight the importance of addressing gaps in PGx research for diverse populations (lines 376-379):

"However, the **post** prescribed and effective dose of simvastatin in Taiwan ranges from 10-20mg³²⁻³⁴. **These observations emphasize the need for expanding PGx studies in underrepresented populations to refine guidelines and improve precision medicine globally.**"

26) I don't think Firstly and thirdly should be used

Response: We have revised the manuscript accordingly to the suggestion.

Here are revised sentences (lines 389-401):

"**First**, while our findings validate the published results that PGx risk variants increase the risk of adverse events, and the increase is statistically significant, the relative risk is low or moderate. **Second**, the vast majority of those with PGx risk variants who take the drug in question do not suffer from the predicted side-effects. Conversely, a significant fraction of those without PGx risk variants suffer from adverse events. This underscores that PGx is not an absolute predictor of ADEs but a critical tool that complements other clinical

factors, such as patient history, comorbidities, and drug-drug interactions, to guide treatment decisions. When used in conjunction with these factors, PGx information can help identify at-risk patients and tailor therapies more effectively. **Third**, many of the adverse events are reversible, non-life-threatening events, and can be managed easily.”

27) Lines 365-370 seems pretty controversial - PGx is to be used as an additional piece of information and not an end all be all - I think it would be good to highlight the utility of PGx information is in addition to other clinical factors

Response: We agree that PGx should be viewed as an additional piece of information rather than an absolute determinant in clinical decision-making. We have revised the text to emphasize the complementary role of PGx alongside other clinical factors.

Here are the revised sentences (lines 394-399):

“Conversely, a significant fraction of those without PGx risk variants suffer from adverse events. **This underscores that PGx is not an absolute predictor of ADEs but a critical tool that complements other clinical factors, such as patient history, comorbidities, and drug-drug interactions, to guide treatment decisions. When used in conjunction with these factors, PGx information can help identify at-risk patients and tailor therapies more effectively.**”

28) Consider providing examples from your patients of the reversible, non-life threatening events (line 367) and times when there is no equally effective drug in many cases (368) and add references to these statements

Response: Per the reviewer’s helpful suggestion, we have included in the revised manuscript specific examples of reversible, non-life-threatening events and cases where no equally effective drug alternatives are available. For reversible events, we added statin-associated myalgia as an example, referencing its mild and reversible nature upon cessation of the drug, with supporting literature. For cases where no equally effective alternatives are available, we provided clopidogrel as an example, particularly for elderly or high-bleeding-risk patients, and cited evidence from the CURE and POPular AGE trials highlighting its safety and efficacy in these populations

Here are revised sentences (lines 399-415)

“Third, many of the adverse events are reversible, non-life-threatening events, and can be managed easily. For example, statin-associated myalgia, characterized by muscle pain or weakness, occurs in up to 15% of treated patients. Most cases are mild and completely resolved upon discontinuation of the statin, with symptoms improving within an average of 2–3 months. In rare cases, rechallenging with a different statin may result in successful tolerance without recurrent symptoms¹⁻³. Finally, as there are no equally effective drug alternatives available in many cases, avoiding adverse events by not taking a drug means that one is taking a drug of lower efficacy. For instance, clopidogrel is a widely used P2Y12 inhibitor in elderly patients with ACS or those undergoing percutaneous intervention (PCI). While alternatives like ticagrelor or prasugrel offer enhanced antiplatelet effects, they are associated with significantly higher risks of bleeding, making clopidogrel the safer and more practical option for patients older than age 70 or with higher bleeding risk⁴⁻⁶. This underscores the clinical challenge of balancing efficacy, safety, and patient-specific factors when managing pharmacogenetic risks.”

Reference:

1. Cholesterol Treatment Trialists, C. Effect of statin therapy on muscle symptoms: an individual participant data meta-analysis of large-scale, randomised, double-blind trials. *Lancet* 400, 832-845 (2022). [https://doi.org/10.1016/S0140-6736\(22\)01545-8](https://doi.org/10.1016/S0140-6736(22)01545-8)
2. Fitchett, D. H., Hegele, R. A. & Verma, S. Cardiology patient page. Statin intolerance. *Circulation* 131, e389-391 (2015). <https://doi.org/10.1161/CIRCULATIONAHA.114.013189>
3. Hansen, K. E., Hildebrand, J. P., Ferguson, E. E. & Stein, J. H. Outcomes in 45 patients with statin-associated myopathy. *Arch Intern Med* 165, 2671-2676 (2005). <https://doi.org/10.1001/archinte.165.22.2671>
4. Gimbel, M. *et al.* Clopidogrel versus ticagrelor or prasugrel in patients aged 70 years or older with non-ST-elevation acute coronary syndrome (POPular AGE): the randomised, open-label, non-inferiority trial. *Lancet* 395, 1374-1381 (2020). [https://doi.org/10.1016/S0140-6736\(20\)30325-1](https://doi.org/10.1016/S0140-6736(20)30325-1)
5. Capranzano, P. & Angiolillo, D. J. Antithrombotic Management of Elderly Patients With Coronary Artery Disease. *JACC Cardiovasc Interv* 14, 723-738 (2021). <https://doi.org/10.1016/j.jcin.2021.01.040>
6. van den Broek, W. W. A. *et al.* Cost-effectiveness of clopidogrel vs. ticagrelor in patients of 70 years or older with non-ST-elevation acute

coronary syndrome. *Eur Heart J Cardiovasc Pharmacother* **9**, 76-84 (2022). <https://doi.org/10.1093/ehjcvp/pvac037>

29) lines 372-380 - please note the CPIC and FDA PGx guidance often does evaluate and provide risk mitigation management strategies

Response: We agree that these resources often provide comprehensive evaluations and recommendations for risk mitigation strategies in PGx-guided therapy. We have revised the manuscript to explicitly acknowledge the contributions of CPIC and FDA guidance in this area and contextualized our findings and recommendations accordingly.

Here are revised sections (lines 417-429):

“Given our findings, one must proceed with caution when implementing PGx-guided therapy. Existing resources, such as the clinical practice recommendations from CPIC and US FDA, already provide valuable recommendations for risk mitigation and management strategies in some scenarios. However, further studies must be conducted to (1) identify additional (genetic and non-genetic) factors that cause ADEs that explain the baseline occurrence of such events in those without PGx risk variants; (2) explore protective factors that allow carriers of PGx risk variants to tolerate drugs without adverse events; and (3) develop and refine risk-management strategies for PGx risk variant carriers, particularly for scenarios where CPIC and US FDA have not yet offer comprehensive recommendations. Such efforts will ensure that individuals can safely benefit from the most effective therapies while minimizing risks.”

30) did you do NGS or WGS - refers to both in the manuscript

Response: In the original manuscript, we used both NGS and WGS inconsistently, which may have caused confusion. To ensure clarity and accuracy, we have revised the manuscript to consistently refer to WGS, as it is the specific sequencing approach utilized in this study. This revision eliminates ambiguity and ensures consistency throughout the manuscript.

31) did you account for phenocoverision?

Response: Although we adjusted for certain factors, such as concurrent medications and comorbidities, that could influence drug response, phenoconversion was not explicitly evaluated. We have now acknowledged this in the discussion section as a limitation and highlighted its potential

impact on the observed genotype-phenotype associations. Future studies should directly account for phenoconversion to improve the precision of pharmacogenetic analyses.

Here are the inserted sentences (lines 463-475):

“Phenoconversion, the alteration of an individual’s observed drug-response phenotype due to external or environmental factors such as concurrent drugs, comorbidities, or lifestyle, is another important consideration¹⁻³. Although we adjusted for comorbidities and concurrent drugs, phenoconversion was not explicitly evaluated. This limitation could potentially confound the observed associations between PGx variants and clinical outcomes. For example, concurrent drugs that inhibit or induce drug-metabolizing enzymes could mask or amplify the effects of PGx variants, leading to misclassification of phenotypes. Future studies should incorporate phenoconversion explicitly by leveraging detailed drug histories and environmental data to better distinguish genetic from non-genetic influences on drug response. Addressing phenoconversion will help refine genotype-phenotype associations and improve the clinical utility of PGx-guided therapy.”

Reference:

1. Klomp, S. D., Manson, M. L., Guchelaar, H. J. & Swen, J. J. Phenoconversion of Cytochrome P450 Metabolism: A Systematic Review. *J Clin Med* 9 (2020). <https://doi.org/10.3390/jcm9092890>
2. den Uil, M. G. et al. Pharmacogenetics and phenoconversion: the influence on side effects experienced by psychiatric patients. *Front Genet* 14, 1249164 (2023). <https://doi.org/10.3389/fgene.2023.1249164>
3. Shah, R. R. & Smith, R. L. Addressing phenoconversion: the Achilles' heel of personalized medicine. *Br J Clin Pharmacol* 79, 222-240 (2015). <https://doi.org/10.1111/bcp.12441>

32) line 436 -should "image" be "imaging"?

Response: As suggested, the sentence has been updated with “imaging” (line 509).

“They included outpatient, inpatient, and emergency room visiting records, drug prescription records, discharge summaries and operation notes, laboratory test results, and reports of pathology, surgery, **imaging**, and Mini-Mental State Examination.”

33) consider adding the weblinks tot he references vs in the text of the manuscript

Response: Upon review, we recognized that some weblinks were placed in the text while others were in the reference list. We have now moved all weblinks to the reference list for consistency and adherence to standard formatting guidelines.

34) add the date accessed to the web links

Response: As suggested, we have included the date of access for the web links.

Here is the revised sentence (lines 551-554):

“The drug-gene pairs were adopted from US FDA Table of Pharmacogenetic Associations and the Clinical Guideline Annotations table of CPIC guidelines from PharmGkB (**accessed on 2024/02/27**).”

35) add references to the CPIC and FDA guidance specific to the drug-gene combinations

Response: We have revised the text and added references accordingly in the Method section (lines 557-579).

“The treatment outcome assessment focused on 4 drug-gene pairs, azathioprine (*NUDT15/TPMT*), clopidogrel (*CYP2C19*), statins (*ABCG2/CYP2C9/SLCO1B1*), and NSAIDs (*CYP2C9*)¹⁻⁵.”

Reference:

1. FDA, U. *Table of Pharmacogenetic Associations*,
<<https://www.fda.gov/medical-devices/precision-medicine/table-pharmacogenetic-associations>>
2. CPIC. *CPIC® Guideline for Thiopurines and TPMT and NUDT15*,
<<https://cpicpgx.org/guidelines/guideline-for-thiopurines-and-tpmt/>>
3. CPIC. *CPIC® guideline for statins and SLCO1B1, ABCG2, and CYP2C9*,
<<https://cpicpgx.org/guidelines/cpic-guideline-for-statins/>>
4. CPIC. *CPIC® Guideline for Clopidogrel and CYP2C19*,
<<https://cpicpgx.org/guidelines/guideline-for-clopidogrel-and-cyp2c19/>>
5. CPIC. *CPIC® Guideline for NSAIDs based on CYP2C9 genotype*,
<<https://cpicpgx.org/guidelines/cpic-guideline-for-nsaids-based-on-cyp2c9->

genotype/>

36) line 500 - how did you measure compliance?

Response: Compliance was assessed using descriptions extracted from outpatient medical records. Since the TPMI medical records contain unstructured physician notes, our collaborating physicians reviewed these records to identify relevant information regarding compliance. When clarification was needed, we consulted with the physicians to confirm the content. We have updated the Methods section for clarification (lines 604-609).

“Compliance was assessed by reviewing descriptions extracted from outpatient medical records. Since the TPMI medical records contain unstructured physician notes, collaborating physicians and pharmacists reviewed these records to identify information regarding compliance. When clarification was needed, we consulted with the physicians to confirm the content.”

37) line 523 - celecoxib does not need to be capitalized

Response: We have revised the text to ensure "celecoxib" is not capitalized.

38) line 525-531 - consider adding more explanation of what the numbers/letters are in the () CPT codes or ?

Response: Thank you for your suggestion. We have added a supplementary table (Supplementary Table 12) detailing the NHIRD order codes used in our study along with their descriptions and relevant notes to enhance clarity.

Additionally, we have revised the manuscript text to include the reference to the NHIRD coding system website:

<https://info.nhi.gov.tw/INAE5000/INAE5001S01>.

Here is the added Supplementary Table 12:

Supplementary Table 12: Descriptions of NHIRD order codes for dialysis and renal replacement therapy procedures

NHIRD Order Code	Name
58001C	Hemodialysis
58002C	Peritoneal dialysis
58009B	Continuous ambulatory peritoneal dialysis · CAPD - 1.CAPD instruction
58010B	Continuous ambulatory peritoneal dialysis, CAPD - 2.CAPD, single unit P.D. set transfer material fee
58011C	Continuous ambulatory peritoneal dialysis, CAPD 3.Peritoneal Dialysis Follow up therapy (1)CAPD
58012B	Continuous ambulatory peritoneal dialysis · CAPD 4.CAPD · Tenckhoff catheter implantation
58013C	Ascites dialytic ultrafiltration
58017C	Continuous ambulatory peritoneal dialysis, CAPD 3.Peritoneal Dialysis Follow up therapy (2)Automated peritoneal dialysis
58018C	Continuous veno-venous hemofiltration dialysis (C.V.V.H.D)
58026C	Home visit - home dialysis therapy
76020B	Renal implantation
N26028	Da Vinci assisted renal implantation

39) line 501 - how did you define several allergic reactions?

Response: Allergic reactions were defined based on descriptions documented in the medical records by the responsible physicians. These included symptoms such as rash, urticaria, or other hypersensitivity reactions attributed to AZA use. We have added this explanation to the Methods section for clarity (lines 617-620).

“Allergic reactions were defined as hypersensitivity events documented by the responsible physicians in the medical records, including symptoms such as rash, urticaria, or other similar reactions attributed to AZA use.”

Reviewer #3 (Remarks to the Author):

In this manuscript Wei et al. leverage data from the large Taiwan Precision Medicine Initiative and report on the overall prevalence of genetic variants known to modulate pharmacogenetic outcomes. The results are interesting for they provide a window into a still under-represented population. They report on the presence of actionable variants in 19 genes that can affect the response of 58 commonly prescribed medications. In addition, derivation of empirical prescription events and drug dosages as exposed in Table 1 can be a valuable resource for future studies on the cost-effectiveness of the implementation of PGx programs within the Taiwanese population.

Response: We appreciate the reviewer’s recognition of the value of our findings while noting the limitations of our study design and scope. We have addressed all the important points raised below and revised the manuscript per the helpful suggestions.

Unfortunately, the reach of their analysis is limited. They were only able to access the association to 4 PGx gene-drug pairs and, despite the very large sample size, by using a retrospective design with data captured through EMR could not derive reliable estimates of the occurrence of side effects or adverse reactions.

Response: We agree that if our aim was to replicate the association of risk of the 4 gene-drug pairs, the reach of our analysis would be limited. However, our aim is NOT to replicate previous finding, but to harness the kind of data state-of-the-art large-scale studies provide to ask the question: if one uses the

PGx guidelines based on widely accepted genetic risk variant alone, what is the clinical impact? The Taiwan electronic medical record is comprehensive (although we could not capture the participants total health data, as noted in the discussion section of the manuscript), allowing us to extract prescription and compliant data based on multiple clinical visits and multiple physician notes detailing adverse drug effects (ADEs) and reason for medication/prescription change. As such, our study precedes those to be reported by the US Our Future Health Project or the US All of Us Project. We focused on only 4 gene-drug pairs where we are confident in the data quality AND have sufficient numbers for statistical power. Our results show that the excess risk of ADE for these widely accepted PGx risk variants is small and does not justify wholesale use of PGx variant status in prescribing drugs in the clinical setting. As such, the reach of our study is universal and provides insight for further refinement of PGx risk-based therapy.

In addition, as would be expected with such a study design, they were not able to access the potential for improved outcomes based on PGx information. In result, I believe the results are noteworthy for a specific audience, but are not necessarily original, for the authors reproduce results already well-known in the literature.

Response: The reviewer is correct that our study was not designed to assess formally the potential for improved outcomes based on PGx information. However, our observations point to limited utility in predicting the outcome of an individual prescribed a medication based on PGx risk variant status. Given our results are based on a large cohort in real life settings, which has never been done before, there is global implication. While there is clearly a difference in incidence of ADE between those with or without a PGx variant in a POPULATION, the predictive value for an individual is low. This implication has implications and should be broadly disseminated.

I also have some technical concerns.

Probably the most unclear section of the manuscript is the extraction and validation of data for Clopidogrel, AZA, statins and NSAIDs (the 4 PGx gene-drug pairs). Each of these cohorts/studies should be detailed and the data extraction process and validation should be thoroughly described.

Response: As suggested, we have expanded the “methods” section to describe the extraction and validation of data for the 4 PGx gene-drug pairs.

In brief, we analyzed the prescription records to define the duration of drug exposure for each gene-drug pair. For patients who discontinued a drug, we carefully examined the corresponding time interval to identify ADEs. To achieve this, we utilized both structured and unstructured EMRs, including hospitalization records, surgical reports, imaging findings, laboratory results, and physician notes. The physician notes were particularly valuable for documenting the occurrence of ADEs and the reasons for discontinuation. Collaborating physicians and pharmacists conducted thorough reviews of these records to validate the reported outcomes. The TPMI team undertook rigorous data cleaning processes to standardize clinical test results and integrate structured data, such as ICD codes and laboratory values, across participating institutions to minimize variability in EMRs from different hospitals.

Here are revised sections (lines 571-648):

“Drug prescription and treatment outcome assessment

When analyzing the number of individuals encountered drugs with PGx warnings, 58 approved drugs with clear genetic-base clinical utility **recommendations** were selected based on their availability in the TPMI dataset. **Individuals with only a single prescription record for each drug were excluded.**

The treatment outcome assessment focused on 4 drug-gene pairs, azathioprine (*NUDT15/TPMT*), clopidogrel (*CYP2C19*), statins (*ABCG2/CYP2C9/SLCO1B1*), and NSAIDs (*CYP2C9*)^{6,21,57-59}. Data from people taking AZA, clopidogrel, statins, and NSAID metabolized by *CYP2C9* were analyzed to assess the influence of PGx variants and phenotypes on drug response. **To assess the impact of pharmacogenomic (PGx) variants on drug response, we analyzed prescription records and treatment outcomes for four gene-drug pairs: azathioprine (*NUDT15/TPMT*), clopidogrel (*CYP2C19*), statins (*ABCG2/CYP2C9/SLCO1B1*), and NSAIDs (*CYP2C9*). Prescription records were examined to define the duration of drug exposure, and ADEs were identified by reviewing corresponding time intervals following drug discontinuation. Both structured and unstructured EMRs were utilized, including hospitalization records, surgical reports, imaging findings, laboratory results, and physician notes. Unstructured physician notes provided valuable insights into ADEs and reasons for drug discontinuation. Collaborating physicians and pharmacists reviewed these records to validate reported outcomes. Data cleaning and standardization processes were conducted by**

the TTPMI team, integrating clinical test results and structured data, such as ICD codes and laboratory values, to reduce variability across participating institutions.

Clopidogrel and CYP2C19:

To study the influence of *CYP2C19* on clopidogrel response, we included clopidogrel users aged ≥ 18 years who demonstrated good compliance and lacked severe allergic reactions. The primary endpoint was MACE, including CV death, non-fatal HF, non-fatal UA, acute MI, acute ischemic stroke or transient ischemic attack, or TLR requiring clinical interventions such as PCI or surgical bypass (extracted from image and operation reports). Compliance was assessed by reviewing descriptions extracted from outpatient medical records. Since the TPMI medical records contain unstructured physician notes, collaborating physicians and pharmacists reviewed these records to identify information regarding compliance. When clarification was needed, we consulted with the physicians to confirm the content.

Azathioprine and NUDT15/TPMT:

The AZA study cohort comprised of AZA tolerant controls and patients who discontinued AZA due to massive hair loss, GI discomfort (nausea, vomiting, and diarrhea), allergic reactions, hepatitis (defined as ALS/AST $> 3x$ ULN), leucopenia (defined as WBC $< 3,500/mm^3$), and thrombocytopenia (defined as platelet count $< 150,000 /uL$), as documented by the responsible physicians and/or laboratory test results. Allergic reactions were defined as hypersensitivity events documented by the responsible physicians in the medical records, including symptoms such as rash, urticaria, or other similar reactions attributed to AZA use. Patients with pre-existing hematological malignancies or poor liver function were excluded. The study design is shown in Supplementary Fig 2.

Statins and ABCG2/CYP2C9/SLCO1B1:

The TPMI participants taking atorvastatin, fluvastatin, lovastatin, pitavastatin, pravastatin, rosuvastatin, or simvastatin without any muscular disorder history were included to assess the role of *ABCG2*, *CYP2C9*, and *SLCO1B1* in the risk of SAMs (CPK elevation with muscle complaints, myalgia, myositis, and rhabdomyolysis).

NSAIDs and CYP2C9:

For NSAID users, we assessed adverse renal events and upper GI discomfort in different **CYP2C9** phenotype groups if they had prescriptions of NSAIDs mainly metabolized via **CYP2C9** (celecoxib, flurbiprofen, ibuprofen, meloxicam, piroxicam, and tenoxicam) but without history of end stage kidney disease (ESRD, **ICD10 code:** N18.5, and N19), poor renal function (<15 ml/min/1.73 m²), receiving dialysis (**NHIRD order code:** 58001C, 58002C, 58009B, 58010B, 58011C, 58012B, 58013C, 58017C, 58018C, and 58026C), renal replacement therapy (**NHIRD order code:** 76020B and N26028 and **ICD10 code:** T86.1 and Z94.0), alcoholism (**ICD10 code:** F10), esophageal varices (**ICD10 code:** I85 and I98.2), Mallory–Weiss syndrome (**ICD10 code:** K22.6), liver cirrhosis (**ICD10 code:** K70, K72–74, and K76), GI tract cancer (**ICD10 code:** C15, C16, and C17), and coagulation defects (**ICD10 code:** D65-D68). Comorbidities included diabetes mellitus (**ICD10 code:** E08-E13), hypercholesterolemia (**ICD10 code:** E78.0-E78.5), hypertension (I10), ischemic heart diseases (**ICD10 code:** I20-I25), HF (**ICD10 code:** I50), and cerebrovascular diseases (**ICD10 code:** I60-I69) were analyzed as well to identify their influence on treatment outcome.”

Because the data is retrospective, it is unclear how the authors are dealing with temporality issues, drug dosages, missing or incomplete information from the EMR, multi-drug prescription, and adherence.

Response: Regarding temporality, we ensured that outcomes were clearly linked to the defined drug exposure periods. For example, in the clopidogrel cohort, major adverse cardiovascular events (MACE) were identified through structured data and were limited to events occurring during the documented period of drug use. This included outcomes requiring significant interventions, such as PCI or bypass surgery, as verified through hospitalization and surgical records.

As for multi-drug prescriptions, we performed multivariable analyses to adjust for the potential influence of these concurrent medications. Similarly, adherence was assessed through outpatient records and corroborated by physician documentation, ensuring that only patients with good compliance were included in the analysis. While specific dosage data were not consistently available across all records, we mitigated this limitation by including only patients with at least two prescription records to minimize the misclassification of transient or single-dose exposures. This limitation is acknowledged in the manuscript, and we aim to incorporate more detailed

dosage data in future studies to strengthen our analyses further.
We have revised the manuscript to include clarifying explanations.

It is also unclear what is the sensitivity and specificity of the codes used for extraction within the different medical systems that participated in the study. The same issues also apply to the extraction of AZA adverse reactions from the EMR. How was this data extracted and what are the positive and negative predictive values for these features?

Response: Although these metrics were not directly calculated in this study, we relied on well-established codes and definitions validated in prior literature and national databases. By integrating these reliable sources and combining them with physician-reviewed data, we ensured that our definitions were robust and consistent.

Inconsistencies and measurement error may account for most of the difference between the observed results and results from other studies. In other words, these very same problems bring into question the accuracy of the most important conclusion of the manuscript on this section, which is that the clinical impact of the PGx information is limited.

Response: We recognize the role of inconsistencies and potential measurement errors can play in retrospective studies and we applied systematic data cleaning, standardized case definitions, and multidisciplinary review processes to mitigate inconsistencies and measurement errors. We have expanded the “Methods” section to provide a more detailed explanation of these processes. The fact that we replicated the associations of other studies with the most stringent case/control selection criteria meant that our approach was sound. We stand by our conclusions that the clinical impact of PGx risk variant data alone is limited and welcome other groups to conduct prospective studies with us to validate these findings.

One important point that was not clear from the manuscript is whether the PGx variants analyzed were genotyped or sequenced. From the Methods section is not clear. In the Data Source section it is stated that samples were genotyped and later imputed. However, in the Pharmacogenomic variants analysis section it is stated that "the SNP genotype and imputation data of passed variants were validated by

NGS, Sanger sequencing, and Sequenom MassARRAY platform". Did the authors really confirmed all actionable alleles by 4 different technologies for 500,000 individuals?? Or this was just a validation of the accuracy of the genotyped and imputed variants within a limited number of samples? Please clarify.

Response: We apologize for the confusion. All 486,956 participants were genotypes with the custom designed TPM1 and TPM2 SNP arrays and the PGx variant status of all individuals was based on genotyping results. The validity of the genotyping platform was based on whole genome sequencing of a subset of 1,498 participants. The validity of some PGx variants was based on Sanger sequencing and Sequenom MassARRAY data derived from a representative subset of participants. We have revised the Methods section to clarify this issue, as seen below (lines 543-569).

“Pharmacogenomic variants analysis

The pharmacogenomic variants analyzed in this study were primarily obtained through genotyping performed using custom-designed TPM1 SNP arrays, TPMv1 (686,463 SNPs) and TPMv2 (743,227 SNPs). These arrays were tailored for the Han Chinese population to optimize coverage for GWAS studies and pharmacogenetically relevant markers. Imputation was then performed using the TWB WGS dataset as the reference panel. This process utilized advanced phasing and imputation tools, SHAPEIT5 and IMPUTE5, ensuring robust data quality and coverage. The drug-gene pairs were adopted from US FDA Table of Pharmacogenetic Associations⁶ and the Clinical Guideline Annotations table of CPIC guidelines from PharmGKB⁵⁴ (accessed on 2024/02/27). The drugs with fewer than 10 prescription records were removed from the analysis. The curation of PGX variants and actionable PGx phenotype were based on the instructions from PharmGKB and PharmVar. The PGx variants were first screened with WGS data to remove the ones with allele frequency lower than 0.1%. To further ensure the accuracy of key actionable PGx variants, we conducted a validation process on a representative subset of samples. This targeted validation utilized multiple technologies, including WGS, Sanger sequencing, and the Sequenom MassARRAY platform. This step was not performed for the entire cohort of 486,956 participants but focused on confirming the reliability of genotype calls and imputed variants for clinically actionable alleles. The variants with sensitivity and specificity higher than 99% were saved for further analysis. This multi-layered approach of genotyping, imputation, and targeted validation

was designed to enhance data reliability while balancing the scale of the study. The final list of drug-gene pairs evaluated in this study is found in Supplemental table 2, and the variants information in Supplemental table 1.”

Another minor point is that authors should also consider deriving estimates for models that assume a recessive model for some of the PGx variants studied. It is well described that for some of the variants the effect estimates among different genotypes are preponderant at the homozygous state.

Response: Our analyses separate those due to a dominant model (heterozygous for the risk variant) and to a recessive model (homozygous for the risk variant, usually designated as poor metabolizer (PM); and heterozygous for the risk variants, usually designated as intermediate metabolizer (IM)), but the results were tabulated together, as is standard in the field.

REVIEWER COMMENTS

Reviewer #1 (Remarks to the Author):

This is a very relevant population pharmacogenetics study, which comprises more than 450000 individuals, 19 pharmacogenes, and around 3000 PGx markers in a non-European population. Moreover, four gene-drug pairs were more extensively analyzed in this Han Chinese cohort. It is well-written, and the analyses are very interesting and new. The methods support the conclusions, and the data are robust enough to meet the expected standards and to be reproduced in other cohorts. The answers to the review concerns and the corrections in the manuscript are fine and have improved substantially the document.

Response: We thank the reviewer's recognition of the relevance and the novel analyses of our study.

Reviewer #2:

Thank you for revising and resubmitting

Response: We thank the reviewer for recognizing the importance of this work and providing many helpful suggestions. We have revised the paper as suggested.

Abstract:

states A key advantage of PGx-guided therapy is the avoidance of adverse events

"avoidance" is a strong word. I think it is more appropriate to state One of the advantages of PGx-guided therapy is to decrease the likelihood of adverse events.

Response: We have revised the text to replace " the avoidance" with " to decrease the likelihood" to statement more appropriate. Here is the revised sentence (lines 43):

"A key advantage of PGx-guided therapy is to decrease the likelihood of adverse events."

I am not sure what "repositioning" means here

PGx also gains increasing attention in pharmaceutical industry for its potential in drug

development and drug repositioning4 .

Response: We have revised the phrase “drug repositioning” to “drug repurposing” to better align with the more widely accepted terminology in the pharmaceutical industry. Here is the revised sentence (lines 71):

“PGx also gains increasing attention in pharmaceutical industry for its potential in drug development and drug **repurposing**.”

Intro:

~10 - remove symbol and add text

Response: We have replaced symbol with text. Here is the revised sentence (lines 81):

“In a comprehensive study examining PGx variation within the UK Biobank, it was found that the average participant carried genetic variants that would affect their response to **around** 10 drugs based on CPIC guidelines.”

These sentences seem out of order

Due to limitations in CYP2D6 genotyping, as discussed below, several important gene-drug pairs could not be analyzed. By focusing on these pairs, we ensured both clinical relevance and robust methodological design

Response: To improve the clarity and flow of the sentence, we have revised it as follows (line 108-111):

“To ensure both clinical relevance and robust methodological design, we focused on gene-drug pairs that could be reliably analyzed. As discussed below, limitations in CYP2D6 genotyping meant that several important pairs could not be analyzed and were excluded from the study.”

Results:

Figure 1a does not have X and y titles but 1b does - make consistent

Response: We have added X and Y titles to figure 1a. Here is the modified figure:

Define what the actual numbers/% are here vs "most"

However, based on the available medical data, most patients did not suffer from any of the predicted adverse events

Response: We have revised the sentence to include supporting data to improve clarity.

“However, based on the available medical data, most patients did not suffer from any of the predicted adverse events. **For instance, 85.9% of CYP2C19 LoF allele carriers tolerated clopidogrel without MACE, 78–83% of NUDT15/TPMT risk allele carriers tolerated azathioprine, and over 98% of individuals with high-risk statin-related PGx profiles did not develop muscle-related adverse events.**”

Consider changing non-LoF alleles to no LoF alleles

Patients taking clopidogrel with either one or two CYP2C19 LoF alleles were significantly associated with MACE compared to those with non-LoF alleles (P = 2.97 × 10⁻²⁷, OR = 1.53, 95% CI = 1.42–1.65)

Response: We have changed the term “non-LoF alleles” to “no LoF alleles”. Here is the revised sentence:

“Patients taking clopidogrel with either one or two CYP2C19 LoF alleles were significantly associated with MACE compared to those with **no** LoF alleles (P = 2.97 × 10⁻²⁷, OR = 1.53, 95% CI = 1.42–1.65)”

EM vs NM - there are still some references to EM

NUDT15 EM and 17.8% of TPMT EM suffered from AZA-induced ADEs

Response: We have revised all the “EM” to “NM” in the revised manuscript. Here is the revised sentence:

“Again, the clinical impact was limited, with only 21.4% of NUDT15 IM/PM and

16.7% of *TPMT* IM having severe AZA-induced ADEs, whereas 16.9% of *NUDT15* NM and 17.8% of *TPMT* NM suffered from AZA-induced ADEs.”

Methods

should this be TPM1 for the array? Seems to be referred to differently throughout

TPM and TPM2 array, genome501 wide imputation data, and HLA imputation data. TPM (686,463 SNPs) and TPM2 (743,227 SNPs) arrays are designed by TPMI and Thermo Fisher Scientific for the TPMI project, specifically for the Han Chinese population with superior coverage for GWAS-grid and previously published health related variants.

Response: Thank you for pointing this out. We would like to clarify that the official names of the arrays used in this study are TPM and TPM2, as designated by the Taiwan Precision Medicine Initiative (TPMI) in collaboration with Thermo Fisher Scientific.

Reviewer #3 (Remarks to the Author):

Wei et al. carefully provide answers and more detail into their work. I appreciate the comprehensive answers and the work put into the revision of the manuscript. Their responses are clear and acknowledge all the limitations raised in the first revision. As a result, their answers highlight the problems involved with such large-scale analysis and the difficult mission to quantify the overall and specific effects of pharmacogenetic markers. The team should be commended on such enormous amount of work. However, the same facts also highlight the limitation in deriving the effect estimates that are provided. In particular, my main reservation is not regarding their conclusion, but with the novelty of their findings. The question of whether "if one uses the PGx guidelines based on widely accepted genetic risk variant alone, what is the clinical impact?" has been approached by others previously, with similar results overall. I should point, however, the interpretation of the observed results still presents itself with great heterogeneity within the community (while some authors concentrate on the individual-level predictive effect, others are more preoccupied with the sample or population cost-effectiveness of the approach).

Of course, this does not imply on the correctness of the used approach (which I agree was as good as possible considering the scale, scope, and resources available).

Response: We agree with the reviewer's assessment that our study is not the first (or the last) attempt to address the effectiveness of using PGx guidelines in clinical practice. However, we maintain that the "novelty" of our work is the large cohort size in a "real world setting", which reviewers 1 & 2 deemed a strength of our study. We have pointed out the limitations per reviewer 3's suggestion and cited the other studies, so there is no claim of being the first to study the question, just that ours is the largest study of non-European ancestry to address the limitations of PGx-guided therapy in the clinic.